Article 

# Somatic gene delivery faithfully recapitulates a molecular spectrum of high-risk sarcomas

Roland Imle [1,2,3,4,5,6], Daniel Blösel[1,2,3,6], Felix K. F. Kommoss[1,2,3,7], Sara Placke[1,2,3,6], Eric Stutheit-Zhao[2,3,8,9], Christina Blume[9,10,11], Dmitry Lupar[1,2,3], Lukas Schmitt[1,2,3], Claudia Winter[1,2,3], Lena Wagner[1,2,3], Malte von Eicke[1,2,3], Hannah Walzer[12], Julia Förderer[12], Stephanie Laier[12], Michael Hertwig[13], Heike Peterziel [2,3,9,14], Ina Oehme [2,3,9,14], Sophia Scheuerman [2,3,4,14,15], Christian M. Seitz [2,3,4,14], Florian H. Geyer[2,3,9,16,17], Florencia Cidre-Aranaz [2,3,9,16], Thomas G. P. Grünewald [2,3,7,9,16], Christian Vokuhl[18], Priya Chudasama[9,19], Claudia Scholl [3,9,20], Claudia Schmidt[21], Patrick Günther[5], Martin Sill [2,3,8,9], Kevin B. Jones [22,23], Stefan M. Pfister [2,3,8,9], Robert J. Autry [2,3,8,9] ✉ & Ana Banito [1,2,3] ✉

A major challenge hampering therapeutic advancements for high-risk sarcoma patients is the broad spectrum of molecularly distinct sarcoma types and the corresponding lack of suitable model systems. Here we describe the development of a genetically-controlled, yet versatile mouse modeling platform allowing delivery of different genetic lesions by muscle electroporation (EPO) in wildtype mice. This EPO-GEMM (EPO-based genetically engineered mouse model) platform allows the generation of ten genetically distinct sarcomas on an isogenic background, including the first model of *ETV6::NTRK3*-driven sarcoma. Comprehensive histological and molecular profiling reveals that this mouse sarcoma cohort recapitulates a spectrum of molecularly diverse sarcomas with gene fusions acting as major determinants of sarcoma biology. Integrative cross-species analyses show faithful recapitulation of human sarcoma subtypes, including expression of relevant immunotherapy targets. Comparison of syngeneic allografting methods enables reliable preservation and scalability of sarcoma-EPO-GEMMs for preclinical treatment trials, such as NTRK inhibitor therapy in an immunocompetent background.

Sarcomas are a group of mesenchymal cancers arising in soft tissues or bone that disproportionately affect children, adolescents, and young adults[1]. While sarcomas arising from bone mostly fall into the categories of Osteosarcoma (OS), Ewing Sarcoma (EwS), and Chondrosarcoma (CS), Soft-tissue Sarcomas (STS) are significantly more diverse, with more than 50 genetically distinct entities and even more subtypes. They are broadly classified into rhabdomyosarcomas (RMS), which retain some skeletal muscle differentiation, and non-rhabdomyosarcomas STS (NRSTS), consisting of all remaining STS entities with diverse subtypes[2,3]. Many sarcoma entities display a low

tumor mutational burden (TMB) and are often driven by dominant fusion oncoproteins involving chromatin-associated regulators and transcription factors[4]. Others are characterized by chromosomal instability and high TMB. Unlike many other cancers, substantial therapeutic and prognostic advances have largely been lacking for sarcomas for several decades across all age groups. This is a result of various factors, including their immense heterogeneity and lack of clinically actionable targets[5]. Clinical management of sarcomas essentially relies on extensive empirical experience in multiagent chemotherapy combined with surgery and irradiation. Although few

targeted therapies or immunotherapies have proven efficacious against sarcomas[6], they exhibit differential responses to cytotoxic agents stemming from unknown entity-specific mechanisms of oncogenesis and progression.

A major bottleneck to achieving long-sought therapeutic advances for sarcoma patients is the lack of adequate model systems for mechanistic and preclinical studies[7]. Modeling sarcoma is particularly challenging. Due to their rarity and heterogeneity, patient tissue is notoriously scarce, making derivative models poorly available. Genetically-engineered mouse models (GEMMs) are difficult to establish, given that the exact cell of origin remains elusive for most entities[8] and widespread expression of many sarcoma drivers results in embryonic lethality[9]. Yet, several sarcoma GEMMs could be established by introducing mutations into the germline[10], often requiring conditional Cre recombinase expression in defined lineages and developmental windows[11]. Germline GEMMs typically exhibit multifocal tumor onset under mixed genetic backgrounds and have generally been impractical for preclinical testing. More recently, an alternative method to deliver CRISPR/Cas9 constructs targeting *Nf1* and *Trp53* via muscle electroporation (EPO) was shown to generate sarcomas that resemble those obtained using conventional modeling techniques[12].

In this work, we optimize and expand the in vivo EPO method to use transposon vectors as a somatic gene delivery approach to murine soft tissue in neonatal and adolescent wild-type mice. We envisioned that the speedy and cell-type-agnostic nature of this technique would allow us to model genetically defined sarcomas with localized onset in the physiological context of a fully competent immune system. We apply this optimized method to study various sarcoma-associated genetic alterations and successfully generate a broad range of more than ten fusion-positive and fusion-negative sarcoma mouse models, including the first *NTRK* fusion-driven sarcoma GEMM. We also provide experimental evidence that *Bcor*, which is inactivated in 15–20% of RMS and across other tumor entities[13], can function as a tumor suppressor in the muscle. The array of models recapitulates diverse tumorigenic mechanisms ranging from rewiring of epigenetic machinery by gene fusions (*SS18::SSX1/2*), to uncoupled tyrosine kinase activation (*ETV6::NTRK3*), to aberrant transcription factors (*ASPSCR1::TFE3*), and multistep tumorigenesis based on *Trp53*-mutation-related genomic instability and second hit mutations (e.g., *Nf1* or *Smarcb1*). After assessing the reliability of EPO-GEMMs by integrative cross-species analysis, we further derive syngeneic allograft models (SAMs) to preserve, share and scale this collection of newly established sarcoma GEMMs. We demonstrate the feasibility of preclinical testing in mouse sarcoma cell lines and immunocompetent sarcoma SAMs using NTRK inhibitor treatment in *ETV6::NTRK3*-driven sarcomas.

## Results

### An optimized protocol allows efficient in vivo genetic manipulation of mouse muscle tissue

To establish an optimized protocol for orthotopic induction of murine STS by EPO, we chose a vector combination conveying overexpression of oncogenic *RAS* and inactivation of *Trp53*, which have been shown to co-occur in several high-risk sarcoma entities, such as embryonal RMS (eRMS) and have previously been validated to efficiently drive sarcoma in mice from the *Rosa26* locus upon Cre-LoxP recombination[14]. We performed survival surgery to expose the thigh muscle in 4–6-week-old mice (referred to as P30 hereafter) and delivered a plasmid mix containing (i) Sleeping Beauty (SB13) transposase, (ii) a transposon vector expressing oncogenic *KRAS* (K), and luciferase/GFP reporter genes (either as an all-in-one vector (KGL) or in separate plasmids (K + L), (iii) a CRISPR/Cas9 vector expressing Cas9 endonuclease and a single-guide RNA (sgRNA) targeting *Trp53* (Supplementary Fig. 1a–c). The procedure was performed bilaterally, and methylene blue dye was

used to demarcate the quadriceps muscles before EPO with two 5 mm plate electrodes. To expand the range of accessible cell types and the developmental window for oncogenic transformation, the EPO procedure was also established in neonatal (P0) animals. Here, plasmid injection and EPO were performed transcutaneously (Supplementary Fig. 1a). These initial EPO attempts were unsuccessful in inducing tumors, underscoring the previously noted low efficiency of muscle transfection[15] (Supplementary Data 1). We therefore optimized a dedicated protocol for somatic muscle engineering to induce sarcomagenesis using in vivo bioluminescence imaging (IVIS) 2 days and 1 week post EPO as a surrogate for transfection efficiency using different EPO conditions (Supplementary Fig. 1d). Optimal EPO conditions were determined as five unilateral pulses of 100 V for 35 ms for P30 and 70 V for P0 animals. Pre-treatment with hyaluronidase (Hyal) to temporarily loosen up the extracellular matrix[16], as well as delivering oncogenes and reporter genes on separate plasmids (K + L versus KGL) further increased transfection efficiency (Supplementary Fig. 1d). The additionally tested resuspension of plasmid DNA in cationic polymer Poly-L-glutamate (Glut)[17] did not show added benefit in this setting (Supplementary Fig. 1d).

This optimized protocol resulted in highly efficient tumorigenesis in 2–3 weeks with 100% penetrance and 100% bilateral tumor onset for the *KRAS/Trp53* model for both P30 and P0 EPO (Supplementary Fig. 1e–g). Resulting tumors displayed sarcoma-like morphology with mostly spindle or mixed spindle/epithelioid features (Supplementary Fig. 1h). Uniformly strong expression of the driving *RAS* oncogene was accompanied by high levels of proliferation marker Ki-67 across all tumors (50–70%). About 70% of tumors showed focal positivity for myogenin, indicating some myogenic differentiation (Supplementary Fig. 1h). As expected, the tumors were positive for the *KRAS* transgene and displayed *Trp53* insertion-deletion mutations (indels) (Supplementary Fig. 1i, j). Surprisingly, the initially present IVIS signal was often lost during tumor progression, despite successful genomic integration of the reporter gene transposons (Supplementary Fig. 2a, b). This was particularly evident when reporter genes were expressed from the same vector as the oncogene (KGL) and were accompanied by significantly reduced tumorigenesis (Supplementary Fig. 2c). This phenomenon suggests immunoediting against reporter genes. Therefore, we switched from the outbred CD1 strain, initially chosen due to large litter sizes and good foster qualities, to the inbred C57BL/6 J strain, which is less susceptible to immunogenicity to Luciferase and GFP[18,19]. Indeed, no signs of reporter gene silencing were observed in C57BL/6 J mice, whereas tumorigenesis efficiency further improved to a mean latency of 14 days (P30) and 21 days (P0) with 100% penetrance and bilateral tumor onset (Supplementary Fig. 2d). Since SB13 transposase has previously been shown to be limited in transposable cargo size, we further adapted the system to PiggyBac transposase, which showed improved tumorigenesis efficiency for the larger KGL vector in P0 mice (Supplementary Fig. 2e, f). In summary, we generated an optimized protocol and set of tools to efficiently deliver transgenes and edit tumor suppressors in murine skeletal muscle tissue.

### A versatile genetic toolbox successfully generates several fusion-driven sarcomas

Given the high efficiency of the established sarcoma EPO-GEMM system, we next aimed to investigate a broader spectrum of human sarcoma drivers toward their tumorigenic potential. For this purpose, we generated a versatile toolbox of transposon and CRISPR vectors with a focus on sarcoma-typical fusion oncogenes, allowing rational combination based on human sarcoma profiling data (Fig. 1a, b; Supplementary Fig. 3a, b). Reporter genes were omitted to avoid any immunogenicity and maximize tumorigenesis efficiency. Since many fusion-driven sarcomas exhibit few or no co-occurring alterations, oncogenes were tested alone (together with an empty sgRNA vector)

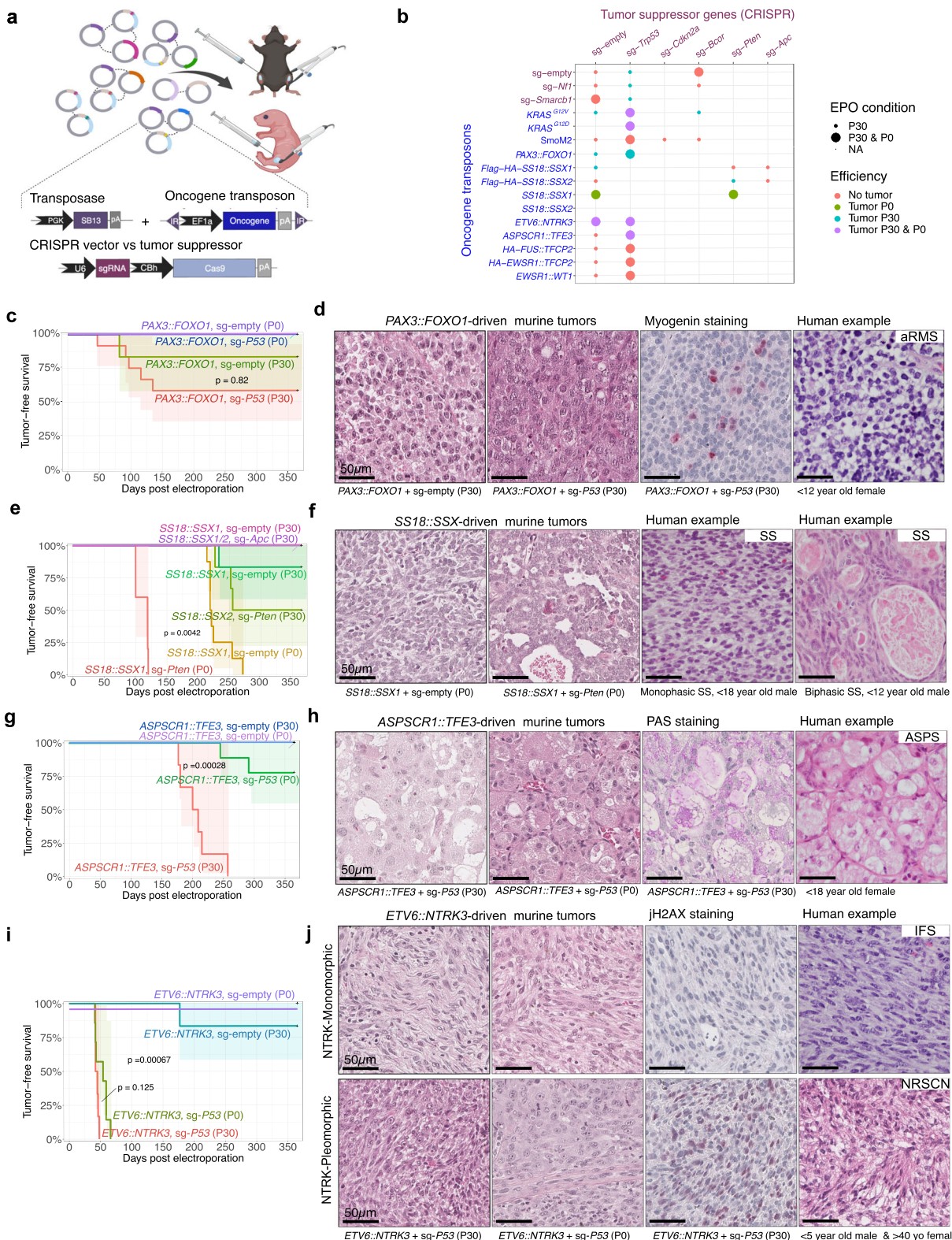

**Fig. 1 | Modeling diverse subtypes of fusion-driven sarcoma. a** Scheme of versatile vector combination for in situ sarcoma induction. Reporter genes were not utilized here. Created in BioRender. Banito, A. (2025) https://BioRender.com/kpnm994. **b** Matrix of vector combinations chosen based on human sarcoma profiling studies. **c** Kaplan−Meier curves of murine *PAX3::FOXO1*-driven tumor induction and representative histographs compared to human aRMS (**d**). **e** Kaplan−Meier curves of murine SS18::SSX1/2 tumor induction and representative histographs compared to human SS (**f**). **g** Kaplan−Meier curves of murine *ASPSCR1::TFE3*-driven tumor induction and representative histographs compared to human ASPS (**h**). **i** Kaplan−Meier curves of murine *ETV6::NTRK3*-driven tumor

induction and representative histographs compared to human IFS and the emerging WHO entity of NTRK-rearranged spindle cell neoplasm (NRSCN) (**j**). Murine tumors exhibited two main histological types, named *ETV6::NTRK3*-Monomorphic and *ETV6::NTRK3*-Pleomorphic. If not specified otherwise, histographs were stained by H&E. All Kaplan−Meier curves correspond to *n* ≥ 6 mice per group electroporated bilaterally. *P* values for comparing Kaplan−Meier curves were determined by two-sided log-rank tests and corrected for multiple testing with the Benjamini-Hochberg method. Scale bars = 50 μm. Source data are provided as a Source data file.

and in combination with sgRNAs targeting tumor suppressors, such as *Trp53* to overcome oncogene-induced stress responses.

A bona fide example of a clinically aggressive fusion-driven STS is alveolar RMS (aRMS), which is mainly driven by the *PAX3/7::FOXO1* fusion gene. In our model system, *PAX3::FOXO1*-induced tumorigenesis was more efficient with concurrent inactivation of *Trp53* compared to *PAX3::FOXO1* alone, which is in line with previous studies showing that inactivation of *Trp53* or *Ink4a/Arf* critically accelerates aRMS development in mice (Fig. 1c; Supplementary Fig. 3c, d)[20]. *PAX3::FOXO1*/ sg*Trp53* tumors occurred unilaterally, with an average tumor-free survival time of 100 days and a 40% penetrance rate. Only one of six mice developed a tumor without simultaneous *Trp53* inactivation. Notably, tumors exclusively emerged in P30 mice, but not in P0 animals. This could be a consequence of the inherently different EPO procedures used for each age group. However, it could also indicate a higher availability of cell types permissive for *PAX3::FOXO1*-driven transformation in P30 mice. This would be in line with earlier observations in conventional GEMMs that maturing rhabdomyoblasts are more susceptible to *PAX3::FOXO1*-driven transformation than embryonic or postnatal muscle stem cells[21]. Histologically, tumors were highly cellular and mitotically active, consisting of sheets of primitive small blue round cells with hyperchromatic nuclei and small nucleoli, akin to histological features observed in human aRMS[2] and conventional mouse models of this disease[20] (Fig. 1d). Unlike human aRMS, Myogenin was partially positive. However, expression of MyoD1 and Myog genes at the mRNA levels was evident in these tumors (Supplementary Fig. 3e).

The most common NRSTS in adolescents and young adults, also occurring in children, is Synovial Sarcoma (SS)[22]. The driving oncofusion *SS18::SSX1* or the less commonly occurring *SS18::SSX2* were electroporated in combination with sgRNAs targeting *Pten* and *Apc*, both of which show occasional loss-of-function mutations in patient specimens. Notably, and unlike *PAX3::FOXO1*-driven tumors, tumorigenesis was significantly more efficient in P0 animals, where even the fusion alone (+ sg-empty) led to tumors with 100% penetrance, albeit with mean latencies of about 230 days and 37% bilateral tumor fraction. Inactivation of *Pten* accelerated tumorigenesis to a mean latency of 112 days, 100% penetrance and 100% bilateral tumor fraction (Fig. 1e; Supplementary Fig. 3c, d). Histologically, tumors were highly reminiscent of SS (Fig. 1f). Tumors generally displayed monophasic differentiation, characterized by cellular sheets and at times fascicular growth of uniform spindle cells with scanty cytoplasm and a delicate "chicken-wire" vasculature. A subset of tumors showed a pronounced collagenous stroma with conspicuous bundles of "wiry" collagen. Notably, 53% of tumors demonstrated areas with epithelial, gland-like structures alongside the spindle cell population, which is the characteristic histological feature of biphasic SS in humans[2]. Anti-HA staining confirmed nuclear expression of *SS18::SSX* in tumor cells (Supplementary Fig. 3f). Overall, *SS18::SSX*-driven EPO-GEMMs were highly reminiscent of human SS and conventional SS mouse models[23], further highlighted by nuclear positivity for TLE-1[24] (Supplementary Fig. 3g).

Another fusion-driven EPO-GEMM model that was successfully established relied on the delivery of the *ASPSCR1::TFE3* oncofusion, characteristic of Alveolar Soft Part Sarcoma (ASPS). Similar to aRMS, tumor induction was dependent on *Trp53* inactivation and was more efficient in mice electroporated at P30 rather than at P0 (100% versus 22% penetrance, 33% versus 0% bilateral tumor fraction, 200 versus 269 days of mean latency) (Fig. 1g; Supplementary Fig. 3c, d). Human ASPS is characterized by its pathognomonic nest-like, alveolar growth pattern with large, polygonal cells with vesicular nuclei and abundant eosinophilic cytoplasm, typically staining positive in periodic acid staining (PAS). This pathognomonic phenotype, reminiscent of pulmonary alveoli, was remarkably well reflected in our model (Fig. 1h), making it indistinguishable from the only hitherto published conventional mouse model[25] and the human disease[2].

Lastly, we also applied our EPO-GEMM system to test gene fusions for which no conventional mouse models have yet been described. A relevant example is *NTRK* translocations, which are considered the characteristic alteration of infantile fibrosarcoma (IFS)[26], but also occur in a broad spectrum of malignancies with diverse tissue origin (e.g., soft tissue, brain, thyroid, breast or uterus)[27]. NTRK inhibitors have recently been approved as entity-agnostic therapeutics, yet suitable models to optimize treatment to prevent or circumvent resistance mechanisms are largely lacking. To bridge this gap, we delivered the *ETV6::NTRK3* fusion gene alone or in combination with a sgRNA targeting *Trp53*, which led to tumors with 100% penetrance in both P30 and P0 animals, with a moderately higher efficiency in P30 mice (45 days versus 52 of mean latency, 100% versus 0% bilateral tumor fraction). Except for one mouse in the P30 group, tumorigenesis was dependent on *Trp53* inactivation (Fig. 1i; Supplementary Fig. 3c, d). Two groups with distinct histological features were observed. 65% (9/14) of tumors showed a fascicular growth pattern consisting of uniform spindle cells with scant cytoplasm and only mild to moderate cytologic atypia. In contrast, 35% (5/14) of tumors showed areas of diffuse growth of spindle to epithelioid cells with a more pronounced eosinophilic cytoplasm and in part (3/5 tumors), severe cytologic atypia. For simplicity, the two groups will be referred to as monomorphic (NTRK-Mono) and pleomorphic (NTRK-Pleo) *ETV6::NTRK* tumors. The first group is reminiscent of IFS histology, while the second resembles a high-grade undifferentiated sarcoma. Tumors from each group did not exhibit differences in *ETV6::NTRK3* mRNA levels (Supplementary Fig. 3h), suggesting instead that a context-dependent role of *ETV6::NTRK3* in different cells of origin or additional acquisition of other genetic alterations in NTRK-Pleo tumors.

Altogether, these results show that the EPO-GEMM approach can be applied to model the tumorigenic potential of several sarcoma-related gene fusions. In each case, the tumors closely recapitulate their human disease counterparts as well as previous conventional GEMMs at the histological level. From all gene fusions tested, only *FUS::TFCP2*, *EWSR1::TFCP2*, and *EWSR1::WT1* did not induce tumors using the current protocol and developmental window. Given the remarkable flexibility of the EPO-GEMM approach, it can be applied to any gene fusion, alone or in combination with relevant sgRNAs, allowing for further optimization of tumorigenesis for different subtypes. The successful generation of a mouse model driven by *ETV6::NTRK3* illustrates the potential of this method for many other alterations that have not yet been modeled in vivo.

## *Bcor* inactivation cooperates with oncogenic *RAS* to drive sarcomagenesis

The flexible somatic gene delivery using the EPO-GEMM system allows probing cooperativity between different sarcoma-related genetic alterations observed in patient tumors with a short turnaround time. Molecular profiling studies have revealed inactivating, truncating or point mutations of the BCL6 corepressor (*BCOR*) gene, encoding a component of the Polycomb repressive complex 1.1 (PRC1.1), in ~15–20% of eRMS and 5% of aRMS[13,28]. Consistent with a role of BCOR in the context of PRC1.1, *BCOR* mutations are concentrated at the C-terminus containing the PUFD (PCGF Ub-like fold discriminator) domain, which is responsible for interaction with the PRC1.1-defining subunit PCGF1 (Supplementary Fig. 4a)[13]. Experimental studies have demonstrated a tumor suppressor function of *Bcor* in subsets of leukemia and medulloblastoma[29,30], but its tumor suppressive role in mesenchymal tumors remains undetermined. We made use of our system to test the hypothesis that *Bcor* inactivation cooperates with oncogenic *RAS* during sarcomagenesis. A sgRNA directed at exon 3 of *Bcor* resulted in increased tumorigenic efficiency (5/8 mice, 62.5%

penetrance) compared to $KRAS^{G12V}$ alone (1/8 mice, 12.5% penetrance). As expected, the effect was not as strong as with *Trp53* inactivation (100% penetrance) (Fig. 2a). Combination with *PAX3::FOXO1* or inactivation of *Bcor* alone did not lead to tumors (0/8 each). *KRAS/sgTrp53* tumors, exhibited focal or diffuse anaplasia, which was often observed in patient specimens of *TP53*-mutated RMS[31]. Although some tumors were positive for desmin and myogenic markers (Supplementary

Fig. 4b), not all showed strong staining as seen in human RMS, and the tumors did not fully recapitulate the histological features of eRMS. *KRAS/sgBcor* tumors were predominantly monomorphic with few signs of pleomorphism and showed focal rhabdomyoblastic differentiation (Fig. 2b). They also exhibited a lower rate of genomic instability and DNA double-strand breaks when compared to *KRAS/sgTrp53* (Fig. 2e–f). In contrast to *KRAS/sgTrp53* tumors, *KRAS/sgBcor* tumors

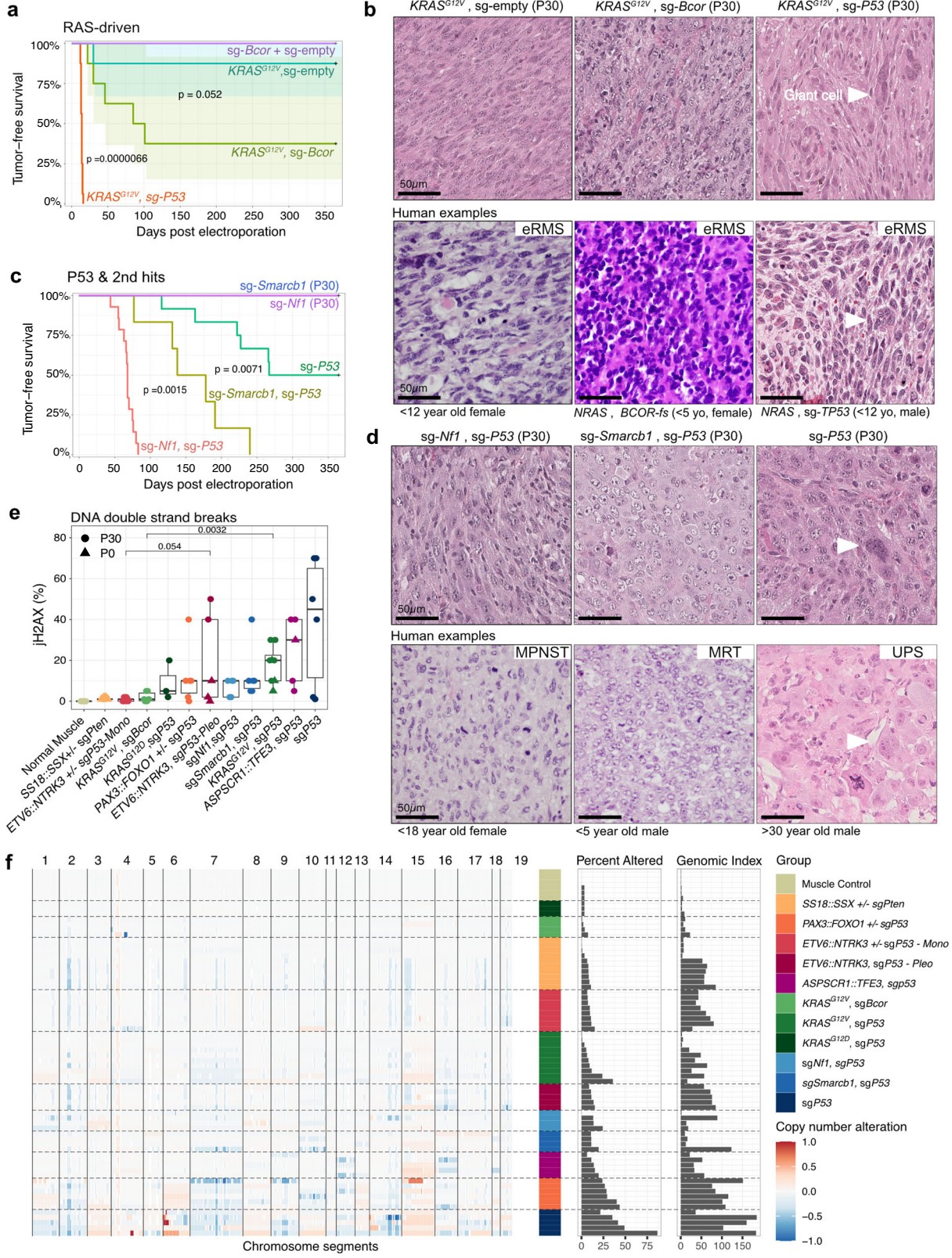

**Fig. 2 | Modeling sarcomas driven by oncogenic RAS and gene inactivation.**
**a** Kaplan−Meier curves of *RAS*-driven mouse sarcomas and representative H&E histographs compared to human eRMS (**b**). **c** Kaplan−Meier curves of mouse sarcomas driven by *Trp53*-inactivation and secondary mutations and **r**epresentative histographs compared to human MPNST, MRT and UPS (**d**). All Kaplan−Meier curves correspond to $n \geq 6$ mice per group electroporated bilaterally. *P* values for comparing Kaplan−Meier curves were determined by two-sided log-rank tests and corrected for multiple testing with the Benjamini-Hochberg method. **e** Integrated score for jH2AX IHC stainings, depicted as boxplots ordered by median from low to high. Boxplots display individual values, median, and interquartile range (IQR). Whiskers extend to the most extreme data points within 1.5 times the IQR from the lower and upper quartiles. *P* values were determined by unpaired two-sided Wilcoxon tests and corrected for multiple testing by the Bonferroni-Holm method. $n \geq 3$ tumors per group. **f** CNV profiles derived from DNA methylation data, condensed as Percent altered ($\leq$/$\geq$. 1) and Genomic Index. $n \geq 3$ tumors per group. Scale bars = 50 μm. Source data are provided as a Source data file.

were negative for mesenchymal markers desmin and myogenin (Supplementary Fig. 4b) and showed a decreased average DNA methylation, possibly indicative of a different cell of origin, differentiation state or epigenetic reprogramming during the transformation process (Supplementary Fig. 4c, d). These results indicate that *KRAS/sgBcor* murine tumors do not correspond to human eRMS. Instead, they suggest that *Bcor* inactivation is not deterministic of one specific sarcoma entity or subtype (e.g., RMS); rather, *Bcor* is a tumor suppressor across multiple contexts. Taken together, these results provide experimental confirmation of *Bcor's* role as a tumor suppressor gene in the muscle, offering a tool to understand its role in tumorigenesis.

### *Trp53* inactivation cooperates with second hit mutations to drive a spectrum of pleomorphic sarcomas

CRISPR-mediated inactivation of *Trp53* alone led to unilateral tumor formation in 50% of cases with a mean latency of 210 days (Fig. 2c), essentially mimicking pleomorphic sarcoma formation in germline-mediated mouse models of Li-Fraumeni syndrome[32,33]. While the models described above each have clearly delineated oncogenic drivers, *Trp53*-mediated genomic instability and diachronous acquisition of second hit mutations over time are the likely cause of oncogenic transformation in this context[34]. This was reflected in pronounced copy number variations (CNVs) (Fig. 2f) and the highest rate of DNA double-strand breaks across the sarcoma EPO-GEMM cohort (Fig. 2e), qualifying this model system to study the nature of mut*Trp53*-mediated genome evolution in the future[35]. Synchronous *Smarcb1* inactivation, pathognomonic for malignant rhabdoid tumors (MRT) and epithelioid sarcomas (EpS), accelerated tumor formation to a mean latency of 173 days with 100% penetrance and 17% bilateral tumor fraction, which is significantly more efficient compared to a previously reported Myf5-Cre-mediated Smarcb1-inactivation model exhibiting 40% and latency longer than 12 months[36]. The efficiency of tumorigenesis was even higher upon synchronous *Nf1* loss, conveying indirect activation of RAS/MAPK signaling (Fig. 2c), typically observed in patients suffering from Neurofibromatosis Type 1 (NF-1), who exhibit an increased risk for malignant peripheral nerve sheath tumors (MPNST) or *Nf1*-deleted eRMS[37]. Combined *Nf1/Trp53* inactivation, yielded tumors with a mean latency of 68 days, 100% penetrance and 86% bilateral tumor fraction, exceeding the efficiency of the previously reported CRISPR-mediated somatic mouse model of combined *Nf1/ Trp53* inactivation, which exhibited a median latency of -100 days[12].

Histologically, sg*Nf1*/sg*Trp53* tumors were mostly compatible with a pleomorphic RMS/spindle cell malignancy with a fascicular growth pattern of spindle or spindle/epithelioid cells (Fig. 2d). The S100 protein neuronal marker was only focally positive in 3/5 tumors, while the myogenic marker myogenin showed stronger focal positivity in 4/5 tumors, suggesting a predominantly myogenic differentiation (Fig. 3a). sg*Smarcb1*/sg*Trp53* tumors consisted of sheets of large epithelioid tumor cells with vesicular and eccentric nuclei amidst an abundant eosinophilic cytoplasm, reminiscent of the characteristic rhabdoid cell morphology of human MRT or EpS (Fig. 2d). While highly positive for mesenchymal marker desmin, they were negative for myogenin and heterogeneous in the expression of other differentiation markers such as ASMA, cytokeratin, and S100 protein (Fig. 3a). sg*Trp53*-only tumors were strongly positive for desmin and showed

heterogeneous expression of other differentiation markers (Fig. 3a). Anaplasia was noted in all three groups, but was most pronounced in sg*Trp53*-only tumors (sg*Trp53* > sg*Smarcb1*/sg*Trp53* > sg*Nf1*/sg*Trp53*) where giant cells and atypical mitoses were also frequently found (Fig. 2d). In conclusion, the sarcoma EPO-GEMM system provides a platform to study *Trp53*-mediated genome evolution and pleomorphic sarcomas driven by tumor suppressor gene inactivation.

### Fusion gene and *Trp53* status determine sarcoma biology and microenvironment

Given the unique opportunity to systematically compare a large number of sarcomas harboring different alterations under identical genetic and experimental backgrounds, we performed an in-depth assessment of their histological phenotypes. A blinded expert pathology review of H&E stains and ten IHC markers was performed across the mouse sarcoma cohort, systematically assessing signs of anaplasia, growth pattern and cellular phenotype as well as immunoreactivity (Fig. 3a, b). While necrosis was generally low except for sg*Smarcb1*/sg*Trp53* tumors, all models were highly proliferative as quantified by Ki-67 staining (Fig. 3c).

Correlation analysis of quantified features (Fig. 3b) confirmed that fusion gene and *Trp53* mutation status are major determinants of sarcoma biology. Compared to non-fusion-driven models, fusion-driven sarcomas typically consisted of rather uniform, spindle cells. In these tumors, occurrences of anaplasia and tumor giant cells were rarely observed. They were also characterized by lower levels of phosphorylated H2AX/jH2AX (DNA double-strand break rates) and reduced immune infiltrates as quantified by CD45 staining. Exceptions were the predominantly small round cell, but likewise uniform growth pattern in *PAX3::FOXO1* tumors and the higher immune infiltrate in tumors driven by *ASPSRC1::TFE1*, also reported in human ASPS[38]. The latter, however, presented some positivity for the S100 marker, which is not commonly observed in ASPS. This could be due to differences in regulation of the S100 cluster in mice and humans, and may also result from common binding sites of MITF and TFE3 genome-wide, as seen for other neural and melanocytic markers that are occasionally positive in ASPS tumors. As observed in human tumors[39], non-fusion-driven models frequently consisted of irregular and spindle to epithelioid tumor cells, frequently exhibiting anaplasia, as well as higher rates of DNA double strand breaks, proliferation and immune infiltration (Fig. 3b–d). Importantly, the positive correlation between genomic stability and immune infiltration found in human sarcoma cohorts was preserved across sarcoma EPO-GEMMs[40]. Although the general degree of immune infiltration was rather low, as expected for sarcomas, leukocyte aggregations up to tertiary lymphocytic structures were occasionally observed (Supplementary Fig. 5a–c).

To further validate the immunophenotype observed by IHC, we employed CibersortX as an algorithm for immune cell deconvolution from bulk RNA sequencing of tumors (Supplementary Fig. 6a, b). There was a modest but statistically significant correlation for the total leukocyte fraction across IHC and CibersortX (Supplementary Fig. 6c), which is likely due to the sampling bias for bulk RNA seq and the overall low immune infiltration observed across all sarcomas. Whereas both *SS18::SSX* and *PAX3::FOXO1*-driven tumors were noticeably immune-cold across both methods and across further immune cell

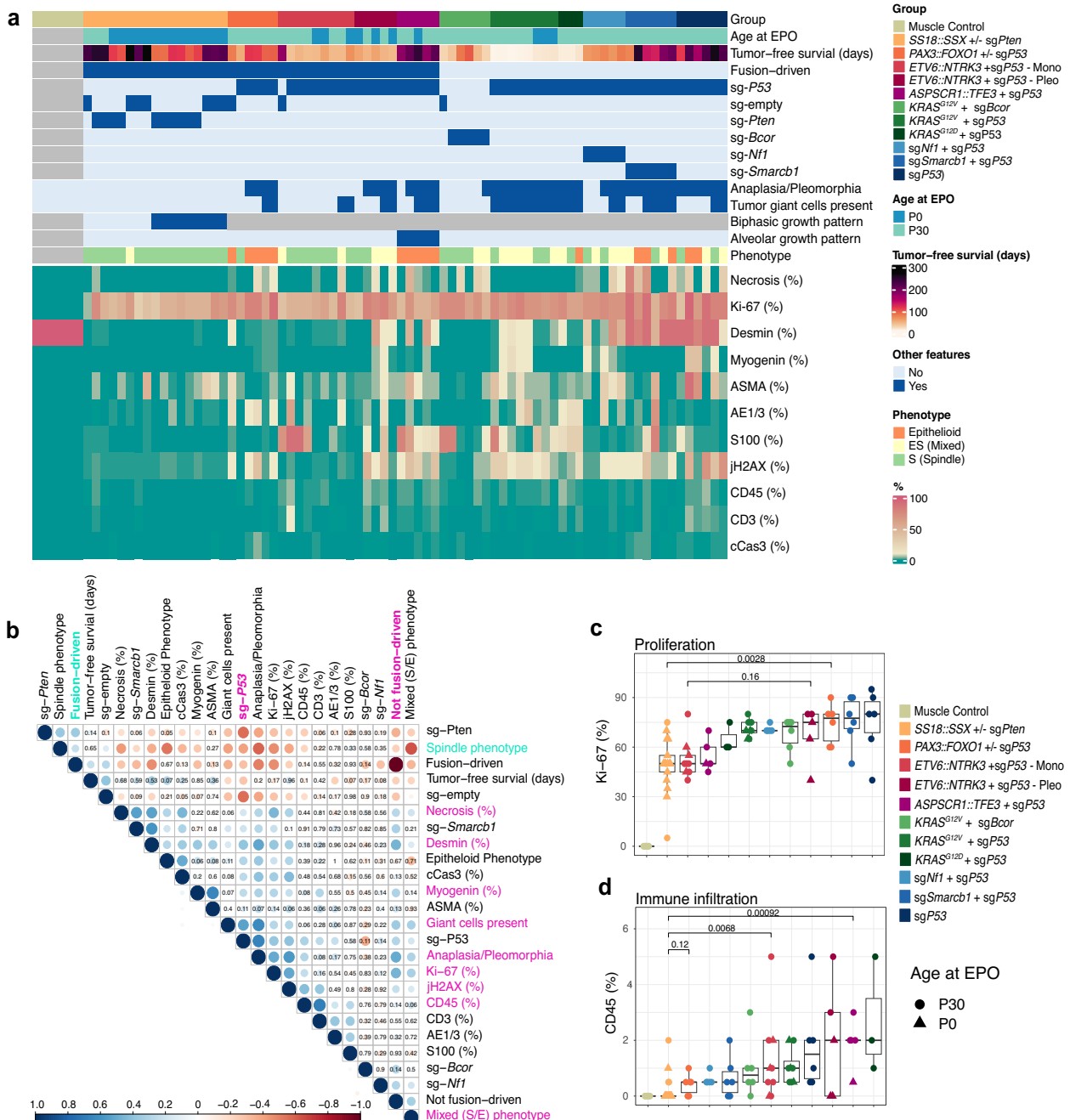

**Fig. 3 | Fusion gene and TP53 mutation status determine sarcoma biology and microenvironment. a** Heatmap view of mouse sarcomas, quantified for six morphological and based on H&E staining and integrated expression scores for 10 antigens determined by immunohistochemistry. Asymmetric color scale for combined visualization of low (CD45, CD3, cCas3) and high-scoring antigens. Quantification by blinded expert pathology review. **b** Pearson correlation matrix based on data from **a**. Insignificant *p* values (≥0.05) are indicated. Blue indicates positive, red indicates negative correlation, and point size reflects effect size. **c**, **d** IHC scores of Ki-67 and CD45 from **a** visualized as boxplots ordered by median from low to high. Boxplots display individual values, median, and interquartile range (IQR). Whiskers extend to the most extreme data points within 1.5 times the IQR from the lower and upper quartiles. *P* values were determined by unpaired two-sided Wilcoxon tests and corrected for multiple testing by Bonferroni–Holm method. *n* ≥ 3 tumors per group. Source data are provided as a Source data file.

subpopulations, RNA-sequencing-based deconvolution failed to detect increased immune infiltration in *ASPSCR1::TFE3* tumors (Supplementary Fig. 6d–g).

Of note, EPO-GEMM primary tumors and derived cell lines, recapitulated the expression patterns of various immunotherapy target antigens (Supplementary Fig. 6a–f), currently explored for human sarcoma treatment[41]. The immune checkpoint surface marker PD-L1 was not upregulated in most tumors when compared to muscle controls, indicating the absence of PD1/PD-L1-mediated immune escape in

these models (Supplementary Fig. 7a)[42,43]. However, several pan-cancer immunotherapy target antigens, including B7-H3/CD276, GD2 and Erbb2/HER2/neu were markedly upregulated in sarcoma GEMMs (Supplementary Fig. 7b–d). Particularly, B7-H3 was strongly upregulated across the entire spectrum of sarcoma EPO-GEMMs (Supplementary Fig. 7b, g), further supporting ongoing clinical approaches of anti-B7-H3 immunotherapy in solid tumors. Additionally, some target antigens were upregulated in subtype-specific fashion, including MCAM (Melanoma cell adhesion molecule) in *ETV6::NTRK3*-Mono or

FGFR4 in *PAX3::FOXO1* tumors, both of which are currently being explored as immunotherapy approaches in Malignant Melanoma[44] and aRMS[45], respectively (Supplementary Fig. 7e, f). Taken together, unsupervised molecular and histological analyses of sarcoma EPO-GEMMs confirmed fusion gene status and *Trp53* mutation status as the major determinants of sarcoma biology.

### Mouse sarcomas exhibit distinct genotype-dependent molecular signatures

Unsupervised tumor classification based on DNA methylation profiling has significantly expanded the means to accurately classify human brain tumors[46] and sarcomas[47]. We adopted this approach to our sarcoma EPO-GEMM cohort by using the recently released 285k mouse methylation array[48]. Clustering of EPO-GEMM tumors based on all or the top 10,000 differentially methylated probes led to genotype-dependent clustering, which was particularly evident for the fusion-driven tumors (Fig. 4a, Supplementary Fig. 4d). Consistent with overlapping microscopic features, non-fusion-driven and *ETV6::NTRK*-Pleo tumors exhibited a more diffuse distribution compared to fusion-driven models. Tumor specimens from a conventional mouse model of SS where *SS18::SSX2* is expressed from the *Rosa26* locus upon Cre recombination[23], group together with SS EPO-GEMMs, indicating conservation of SS18::SSX-mediated biology between these two modeling approaches (Fig. 4a, Supplementary Fig. 4d). One notable phenomenon observed was a broad hypomethylation phenotype in the Synovial Sarcoma group (Supplementary Fig. 4c,d) as previously demonstrated for human SS[39,49]. This is consistent with the hypothesis that the SS18::SSX fusion mediates epigenetic rewiring and oncogenic transformation through binding to unmethylated CpG islands[50]. Tumors from a previously published conventional GEMM of ASPS, expressing *ASPSCR1::TFE3* from the *Rosa26* locus, exhibited distinct DNA methylation profiles (Fig. 4a, Supplementary Fig. 4d), possibly reflecting a different cell of origin, given their strictly heterotopic onset in the cranial vault[25].

Transcriptome-based t-SNE analysis corroborated these results with fusion-driven mouse sarcomas forming distinct groups, each corresponding to genotype-specific expression patterns (Fig. 4b). *KRAS/sgTrp53/sgBcor* and sg*Nf1/sgTrp53* models showed fairly similar transcriptomes, consistent with the commonly underlying upregulation of RAS/MAPK signaling (Fig. 4c). While NTRK-Mono formed a very distinct group, NTRK-Pleo sarcomas clustered in a rather scattered fashion adjacent to *KRAS/sgTrp53* and sg*Trp53* tumors. To further explore the biology underlying induced murine sarcomas, RNA sequencing data were subjected to k-means clustering and gene ontology (GO) analysis (Fig. 4c, Supplementary Data 2). As expected, a group of genes related to skeletal muscle organization and contraction (k-means cluster 4) was clearly upregulated in normal muscle. Upregulation of embryonal developmental pathways (k-means group 1) was shared between various tumor types, while muscle development and differentiation were specifically upregulated in *PAX3::FOXO1* tumors, consistent with mechanistic studies identifying *PAX3/7::FOXO1*-induced activation of myogenic super enhancers[51] (k-means group 7). Consistent with the underlying *NTRK* fusion, signaling pathways of neuron myelination and synapse signaling were specifically upregulated in NTRK-mono tumors (k-means group 6), whereas enrichment of neuron differentiation and multicellular developmental pathways was shared with *SS18::SSX* tumors (k-means group 2). *SS18::SSX*-driven tumors displayed a unique expression signature (k-means group 3), which included developmental transcription factors known to be enriched in human tumors, and genes involved in neuron differentiation, axon development and Wnt signaling[50] (Fig. 4c, Supplementary Data 2). *ASPSCR1::TFE3* tumors shared signatures with other entities and normal muscle, which included upregulation of genes involved in lipid and carbohydrate metabolism (k-means group 5), consistent with findings in a previous conventional mouse model of ASPS[25].

These results demonstrate how the diversity of underlying genetic perturbations drives transcriptomically diverse tumors with a clear genotype-phenotype association. Particularly, sarcoma-driving onco-fusions elicit specific tumor transcriptomes that reflect their respective underlying oncogenic mechanisms.

To systematically assess whether sarcoma EPO-GEMMs recapitulate their human counterparts, we applied a cross-species bioinformatic approach (Fig. 5a). To represent relevant human sarcoma entities and subtypes, we integrated and harmonized RNA sequencing data from the 'The Cancer Genome Atlas' (TCGA) Sarcoma study[39], St. Jude Cloud[52], and the INFORM registry[6], yielding a total of 299 human sarcoma samples that were compared to 63 mouse sarcoma samples. After restricting genes to cross-species-conserved orthologues, the top 2000 most variable genes were batch-corrected for species effects and used for t-distributed stochastic neighborhood embedding (t-SNE) analysis with batch correction for species, dataset and tissue of origin (normal vs tumor). (Fig. 5b, Supplementary Data 3). A similar analysis based on the genes differentially expressed between each subtype versus all other subtypes yielded similar results (Supplementary Fig. 8a, b; Supplementary Data 3). Overall, bona fide fusion-driven sarcomas consistently clustered together with their human counterparts. *KRAS*[G12V] tumors clustered with RMS or a subset of a more heterogeneous tumor group, also containing other sgP53 murine tumors. This group did not overlap with the larger and heterogeneous group of human pleomorphic sarcoma entities, encompassing UPS, MFS and pleomorphic LPS. These results suggest that the variability in expression profiles of human pleomorphic tumors is more difficult to recapitulate in mice using discrete genetic perturbations. It could indicate a myogenic origin of sgP53 murine tumors that is more deterministic in expression-based clustering analysis.

To further compare murine and human tumors, we took advantage of a large dataset of DNA methylation arrays from human sarcomas[43]. Syntenic probe annotation was performed as previously described[48]. Analysis of all 15,218 syntenic probes shared between human and mouse arrays showed a strong species effect for tumorous and normal tissues alike (Supplementary Fig. 8c, d). These findings suggest that the tissues analyzed exhibit divergence in methylation patterns at these regions between the two species. Indeed, prior analysis using this mouse array demonstrated that species-specific effects largely accounted for the observed methylation variation[48]. Selection of probes based on anticorrelation with species ($R^2 > 0.8$) and correlation with class ($p < 0.001$) however, could clearly discriminate mouse and human sarcomas entities and subtypes, with the methylomes from gene fusion sarcomas exhibiting the clearest segregating effects (Supplementary Fig. 8e). These 38 selected probes likely represent regions enriched in highly conserved mesenchymal associated CpGs that may be functionally important in driving subtype specific pathogenesis (Supplementary Data 3).

Altogether, cross-species comparison of transcriptome, methylome, and histology data revealed a coherent picture, indicating conservation of human sarcoma biology across the sarcoma EPO-GEMM cohort, especially for bona fide fusion-driven sarcomas *SS18::SSX*, NTRK-Mono, *PAX3::FOXO1* and *ASPSCR1::TFE3*.

### SAMs preserve GEMMs for long-term application

In terms of therapeutic predictive value, both patient-derived xenografts (PDX) models and GEMMs[53] are regarded as superior when compared to conventional human cancer cell lines, which are still used for ~80% of preclinical therapy trials[53,54]. Most GEMMs have predominantly been used to study the pathobiology of tumors, with limited use in translational applications. One reason for this is the typically mixed genetic backgrounds of conventional GEMMs, which hamper immunocompetent allografting and cross-institutional model sharing. Hence, the broad panel of genetically heterogeneous sarcoma GEMMs established here on a genetically identical C57BL/6J

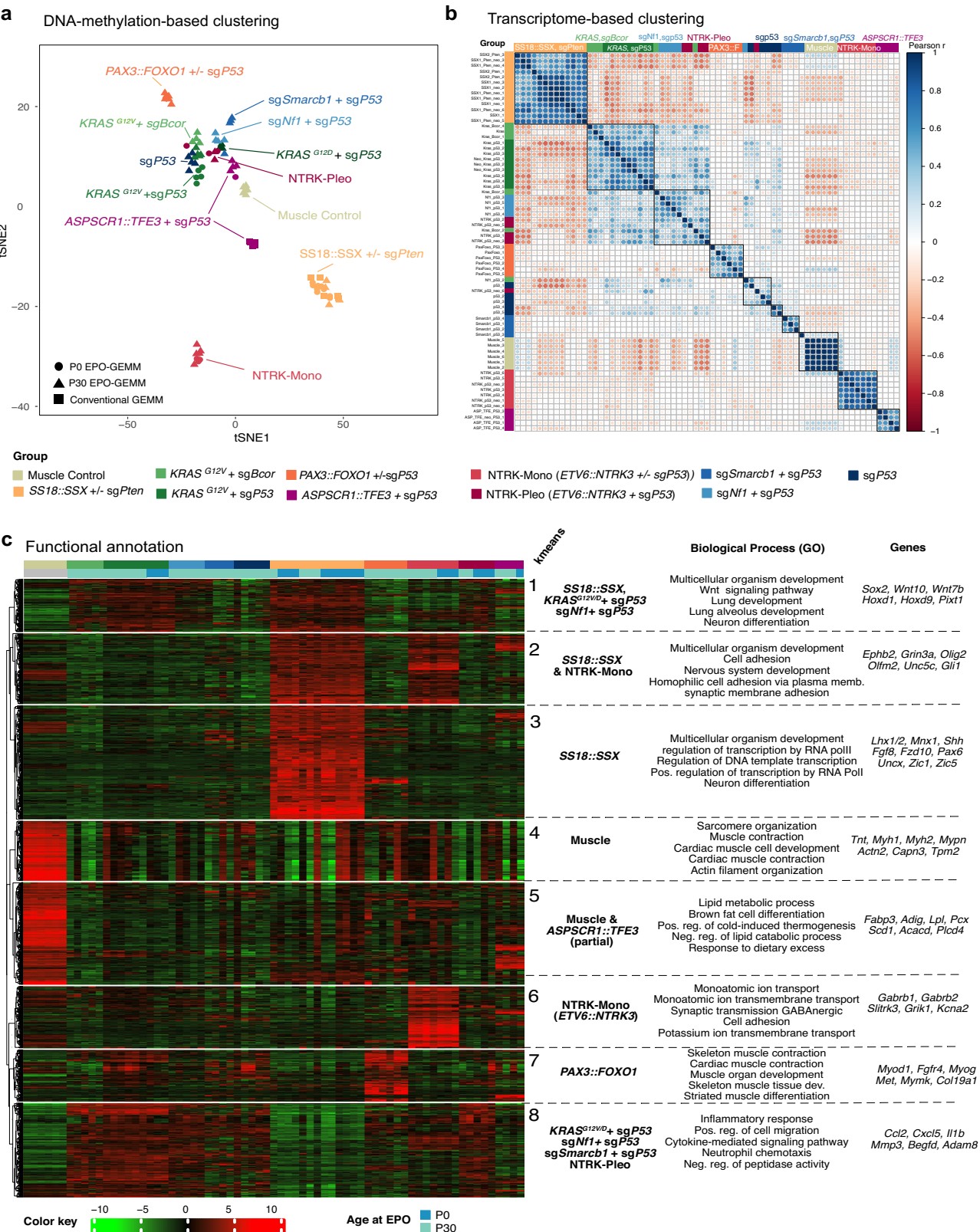

**Fig. 4 | Mouse sarcomas exhibit distinct genotype-dependent molecular signatures. a** tSNE clustering based on the top 10,000 differentially methylated CpG sites. **b** Correlation clustering based on the top 2000 differentially expressed genes. **c** k-means clustering based on the top 2000 differentially expressed genes. The top five most significantly enriched GO terms in the category GOTERM_BP_DIRECT (Biological processes) are shown alongside examples of genes in each k-means group. Source data are provided as a Source data file.

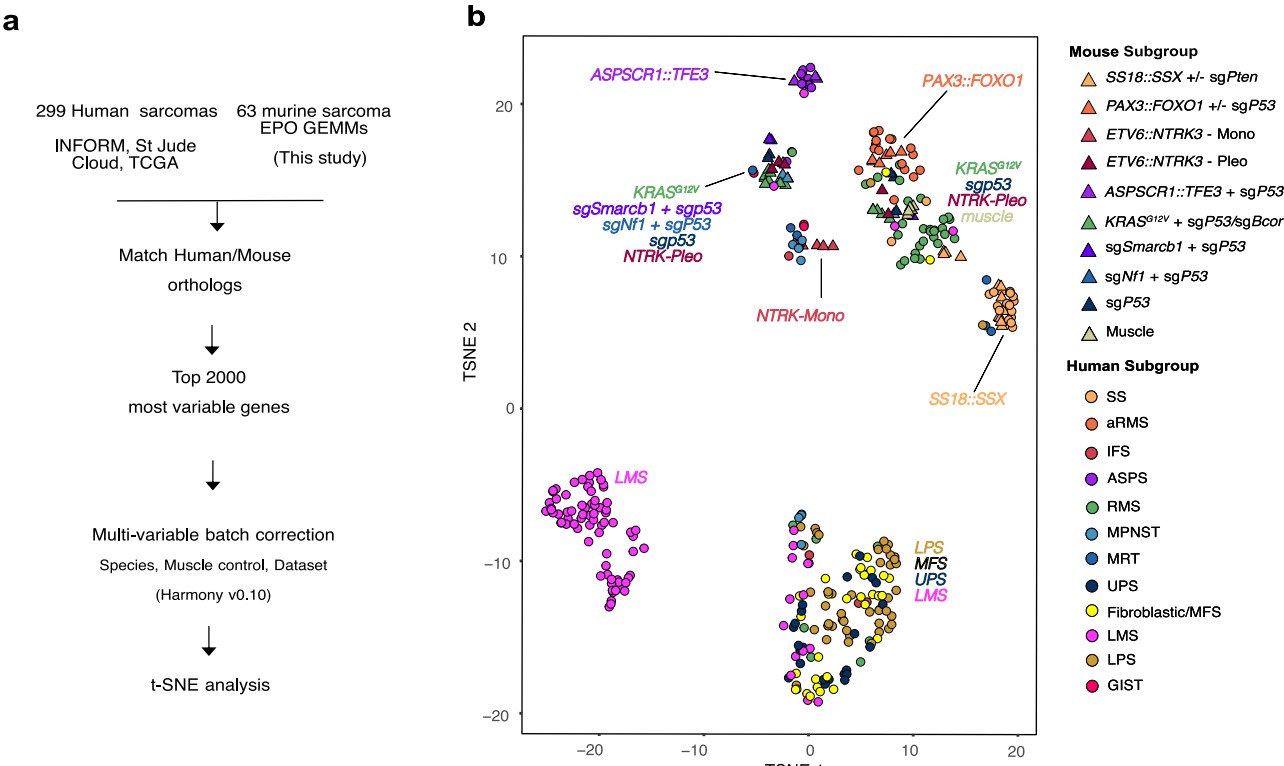

**Fig. 5 | Murine sarcomas resemble a spectrum of human sarcomas. a** Analysis scheme of cross-species sarcoma analysis based on RNA sequencing. **b** tSNE visualization based on cross-species transcriptome analysis of mouse ($n = 63$) and human ($n = 299$) sarcoma specimens. SS synovial sarcoma, aRMS alveolar sarcoma, IFS infantile fibrosarcoma, ASPS alveolar soft part sarcoma, RMS rhabdomyosarcoma, MPNST malignant peripheral nerve sheath tumor, MRT malignant rhabdoid tumor, UPS Undifferentiated pleomorphic sarcoma, MFS myofibroblastic tumor, LMS leiomyosarcoma, LPS liposarcoma, GIST gastro-intestinal stromal tumor. Source data are provided as a Source data file.

background provided an excellent opportunity to explore the appropriate methodology to systematically preserve GEMMs for preclinical treatment trials.

First, we tested whether primary tissue allografts would be superior to *in-vitro*-propagated cells[55] and tumor spheres superior to conventional 2D cultures[56] in preserving the biological properties of the original tumor tissue. Four different SAM types were compared upon orthotopic unilateral engraftment: dissociated primary cells (C), 50−200 μm tumor fragments (F), cells cultured in vitro under 2D conditions (2D) and tumor spheres cultured in vitro under 3D serum-free conditions (3D) (Fig. 6a). All four methods substantially reduced mean TFS from about 78 to 15 days across a variety of genotypes (Fig. 6b; Supplementary Fig. 9a) with engraftment success rates of ~90%, which is significantly higher than in PDX models[53]. Blinded expert pathology review of H&E and IHC sections revealed accurate histomorphology preservation (Supplementary Fig. 9b). While necrosis and proliferation rates were slightly increased, no signs of immune rejection could be observed in any of the four SAM methods tested (Supplementary Fig. 9c–e). Importantly, all SAMs and cell lines clustered together with their respective GEMM upon DNA methylation analysis and showed remarkably high conservation of the underlying methylome features (Fig. 6c, d). Broad hypomethylation in *SS18::SSX* and *PAX3::FOXO1* tumors, for example, was highly preserved across all SAM types, as were CNV profiles (Supplementary Fig. 9f, g). To our surprise, none of the four methods emerged superior in preservation of histotype, genomic stability and DNA methylome, as all models recapitulated the original molecular makeup remarkably well. Even 2D-cultured allografts reflected the distinct histomorphology of fusion-driven sarcoma GEMMs (Fig. 6e). In summary, both primary and cell culture-based allograft methods

were well suited for the preservation of sarcoma EPO-GEMMs for preclinical testing.

Finally, we leveraged the flexibility of in vitro propagation and syngeneic in vivo allografting of sarcoma EPO-GEMMs to apply some of our models to small-molecule testing. NTRK inhibitor therapy with first (Larotrectinib) and second generation (Repotrectinib) agents showed specific and significant activity in NTRK-Mono and NTRK-Pleo cell lines in vitro, comparable to response rates in a rare patient-derived cell model acquired from a patient suffering from NTRK-driven Inflammatory Myofibroblastic Tumor (IMT) (Fig. 6f). Treatment of mice bearing a NTRK-Mono allograft model confirmed high sensitivity to NTRK inhibitors, with Repotrectinib resulting in profound antitumor activity and complete tumor regression (Fig. 6g). These results corroborate the improved potency of second generation NTRK inhibitors and suggest Repotrectinib as a candidate for first line treatment of NTRK3-fusion-driven sarcomas.

## Discussion

The lack of flexible sarcoma models has hindered our understanding of sarcomagenesis and the development and testing of new therapeutic approaches for patients over the last decades. Here we describe an efficient and highly versatile approach to probe any genetic alterations by in vivo somatic engineering of mouse muscle tissue.

We substantially optimized muscle EPO in P30 mice compared to previous studies[12,57], but also adapted the procedure to apply it in neonatal mice, marking the earliest murine soft-tissue EPO to date. Of note, in utero EPO can be applied to brain tissue, but relies on injections into the lumen of cerebral ventricles[58], a structural feature not present in other tissues. In assessing the conditions for the highest transfection efficiency of muscle tissue at two postnatal

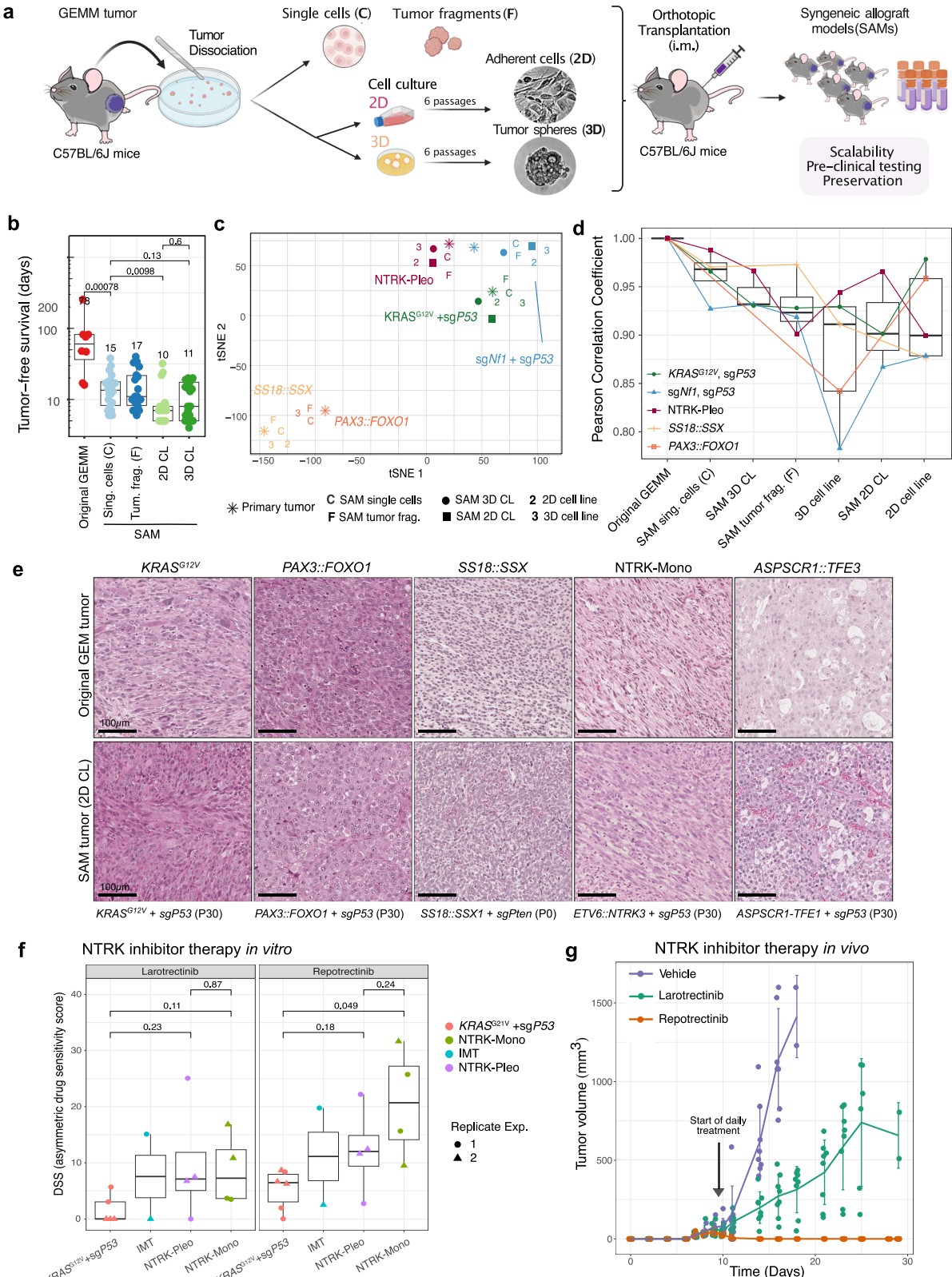

developmental time points, we increased the probability of hitting cells permissive for transformation, even if they are rare. This is particularly relevant for gene-fusion-driven sarcomas, where a precise initial cellular state is thought to be required to allow oncogenic transformation[59]. Indeed, our approach could be applied to several gene fusion sarcomas, each exhibiting a defined DNA methylation profile, most probably primarily reflective of alternative cells of origin.

One could argue that as long as a given cell of origin is present at the time of tissue transfection, the method can be applied to any genetic alteration in the future. Still, there are limitations. Some oncogenes did not give rise to tumors within the observation period of one year post-EPO. These included *TFCP2* fusions (*FUS::TFCP2* and *EWSR1::TFCP2*) and *EWSR1::WT1*, alone or in combination with *Trp53* inactivation. Given that sg*Trp53* alone eventually led to tumors with about 50%

**Fig. 6 | Syngeneic allograft models enable scalability for in vivo testing.**
**a** Schematics of syngeneic allograft modeling (SAM) procedure, systematically comparing four different allograft types. Created in BioRender. Banito, A. (2025) https://BioRender.com/abys0b6. **b** Tumor-free survival of SAMs compared to corresponding GEMMs. $n \geq 8$ tumors per condition, including tumors with different genotypes. **c** tSNE analysis based on the top 10,000 differentially methylated CpG sites of SAMs, mouse-tumor-derived cell lines and corresponding GEMMs. **d** Median-ordered boxplots of Pearson correlation coefficients based on DNA methylation data from GEMMs and corresponding SAMs. $n = 5$ tumors per condition with 5 different genotypes. **e** Representative H&E histographs of GEMMs and corresponding SAMs after orthotopic engraftment of 2D-cultured mouse tumor lines. Scale bars equal 100 μm. **f** NTRKi in vitro sensitivity testing. Data points correspond to asymmetric drug sensitivity scores (DSS, as explained in the methods). DDS = 0 indicates resistance, and values > 10 indicate sensitivity. $n \geq 2$ cell lines grouped by entity, in two independent experiments (triangles and squares). For the IMT condition, only one cell line was available. IMT refers to a human tumor cell line derived from an Inflammatory myofibroblastic tumor driven by *ETV6::NTRK3* used to clone the mouse transposon vectors for electroporation. Three eRMS (*KRAS^G12V*/sg*Trp53*) tumor lines were used as fusion negative controls for comparison. Boxplots in **b**, **d**, and **f** display individual values, median, inter-quartile range (IQR). Whiskers extend to the most extreme data points within 1.5 times the IQR from the lower and upper quartiles. *P* values refer to unpaired two-sided t-tests, corrected for multiple testing using the Bonferroni-Holm method. **g** In vivo treatment of FD-IFS tumor line re-engrafted orthotopically into wildtype C57BL6/J host mice. Treatment started when mice developed palpable tumors (days 8–10 post engraftment) by oral gavage of NTRK inhibitors or vehicle twice a day. Mean, standard deviation and individual values are depicted. $n = 8$ mice per group. Source data are provided as a Source data file.

penetrance, one would expect similar rates when combining sg*Trp53* with these oncogenes. However, the actually observed rates for sg*Trp53* tumors in combination were significantly lower (*TFCP2*-fusions: 2/24 (8%); *EWSR1::WT1* fusion: 1/12 (8%)), indicating that most of the transfected cells were probably cleared due to oncogene-induced stress, cell death or immune-related responses. Precisely why these alterations were unsuccessful in generating tumors could be due to a variety of reasons. One of the most important, certainly, is the availability of permissive cells of origin. DSRCT, driven by *EWSR1::WT1*, presents mostly in the abdomen, possibly originating from mesothelial[60] or other non-myogenic cells[61]. Similarly, TFCP2-fused RMS typically occur intraosseously or are bone-associated, suggesting a non-myogenic origin[62]. For some entity-specific alterations, embryonic cell states may be required for oncogenic transformation, as previously described for the induction of brain tumors by in utero electroporation[58]. This likely explains why *Smarcb1*-inactivation alone was not sufficient for tumorigenesis in our model system, and con-comitant *Trp53*-inactivation was required to drive tumorigenesis[11,36]. For some subtypes, one cannot rule out the need to co-deliver addi-tional alterations to enable tumorigenesis or the requirement to fine-tune oncogene expression levels by the use of strength-variable or endogenous promoters. Finally, some oncogenes may never work simply due to functional divergence between human and mouse genomes. Nevertheless, the fact that it was successful for most entities, including the first *ETV6::NTRK3*-driven model and orthotopic *ASPSCR1::TFE3* model outside the cranial vault[25], clearly suggests that this method is applicable to many other sarcoma subtypes.

Besides the dramatic reduction in animal numbers by avoiding extensive intercrossing, there are several advantages to EPO-GEMMs when compared to previously derived sarcoma models. Perhaps one of the most significant is that it bypasses the need for conditional trans-genic lines and therefore a predetermined suitable lineage or cellular background for transformation. While Cre-transgenic mouse lines have been elegantly used to demonstrate diverse outcomes when certain alterations are expressed in different cellular backgrounds[9,63–65] EPO-GEMMs—or alternatively, the delivery of exogenous Cre (e.g., TAT-Cre injection)—enable targeting of any cell of interest within a given tissue, provided transfection efficiency is sufficiently high. Nonetheless, it is important to emphasize that germline and lineage-specific approaches remain essential tools for investigating the role of cellular context in sarcomagenesis and that the EPO-GEMM platform is intended to complement, not replace, these established strategies. It is also important to note that the same genetic alteration could give rise to tumors with distinct histologic and molecular characteristics (e.g., *ETV6::NTRK3*-driven tumors). This heterogeneity suggests that, for certain oncogenic events, multiple cellular contexts may be permissive to transformation; however, not all resulting tumors will faithfully recapitulate the human disease. As such, histological and cross-species comparisons are critical to confirm that an EPO-GEMM—or any given model—accurately reflects its human counterpart.

The procedure also allows multiplexing different genetic alterations and can be applied to mice of any genetic background. An obvious application is to test cooperation between genetic events. Our approach was able to show that *Bcor* inactivation enables *KRAS^G12V*-driven tumorigenesis in the muscle. The tumors exhibited reduced expression of myogenic markers, and lower overall DNA methylation when compared to *KRAS^G12V*/sg*Trp53* tumors, arguing in favor of a *Bcor* inactivated-specific phenotype related to the induc-tion of permissiveness in alternative cells of origin and/or to an active epigenetic reprogramming that favors transformation. Further stu-dies are needed to determine which muscle-resident cells are tar-geted by EPO and to mechanistically elucidate the role of BCOR loss of function in sarcoma and other pediatric tumors, where it is fre-quently observed[28].

Overall, this flexible approach allowed us to model a very het-erogeneous and diverse spectrum of sarcomas. It generated a geneti-cally diverse set of subtypes that are clinically relevant and span from fusion-driven uniform and immune cold sarcomas to more pleio-morphic, genetically unstable tumors, with a tendency towards increased immune infiltrates, dominated by *TP53* inactivation. Most importantly, the somatically induced sarcoma models were faithful representatives of previously established conventional GEMMs and the human disease. Particularly for fusion-driven sarcomas, these models faithfully recapitulated entity-specific histologies and acti-vated transcriptional programs that closely mirror the biology of the human sarcomas. Another key goal of this study was to preserve the variety of new immunocompetent mouse models for use in preclinical treatment studies. Re-engraftment of GEMM tumor material into syn-geneic C57BL/6 J mice was very efficient without signs of immune rejection. Systematic comparison of four different model preservation methods demonstrated that both primary as well as cell culture-based allografting were suitable for EPO-GEMM preservation and expansion. It should be noted, however, that we did not perform a thorough analysis of immune infiltration profiles in these syngeneic models, and previous studies showed distinct immune landscapes in auto-chthonous sarcomas and their corresponding transplants for one specific subtype[66]. Still, syngeneic models are critical not only for preservation but also to allow the scalability required for preclinical studies. A priority for further application of the models will be in-depth characterization of the immune microenvironments across different genotypes, both in primary and allografted tumors. Lastly, it is important to note that no macroscopic metastases were observed in any of the models described in this study. This absence may be attributed to the limited time frame, as the mice were sacrificed before metastasis could fully develop. Future studies could address this lim-itation by incorporating luciferase-based imaging to detect micro-metastases and manipulating metastasis-promoting genes.

In summary, our study meets a crucial gap in sarcoma and solid tumor research. Not only does it present a suite of models ready for immediate application in both basic and translational research, but it

also introduces an approach that unlocks vast possibilities for a deeper comprehension of sarcoma biology.

## Methods

All animal experiments conducted in this study were carefully planned and approved by the local veterinary authorities and the Regierungspräsidium Karlsruhe, Baden-Wurttemberg, Germany as the responsible authority (animal permits G-36/19, G-2/20, G-3/20). The study adhered to the ARRIVE guidelines, European Community and GV-SOLAS recommendations (86/609/EEC), and United Kingdom Coordinating Committee on Cancer Research (UKCCCR) guidelines for the welfare and use of animals in cancer research. Conscientious application of the 3 R guideline (replacement, reduction, refinement) was emphasized, prioritizing the reduction of potential suffering for the animals.

### Tissue culture and GEM model preservation

Human Embryonic Kidney HEK293T cells (RRID:CVCL 0063), human alveolar RMS cell line Rh30 (RRID:CVCL_0041) and murine neuroblastoma cells N2A (CCL-131) were purchased from the American Type Culture Collection (ATCC) and maintained in DMEM (Gibco) supplemented with 10% Fetal Bovine Serum and 1% Penicillin/Streptomycin (P/S). IMT_NTRK1/INF_R_153 carrying *ETV6::NTRK3* was generated from a primary tumor biopsy obtained from an IMT enrolled in the INFORM (INdividualized Therapy FOr Relapsed Malignancies in Childhood) registry study and cultured in RPMI (Gibco) + 10% FCS + 1% MEM (minimal essential amino acids) + 1% P/S. Human specimens are shown as examples of histological sections in Figs. 1d, f, h, j and 2b, d are part of the Cooperative Weichteilsarkom Studiengruppe (CWS) study group. Written informed consent was obtained from patients or their legally authorized representatives for both the INFORM and CWS registries.

For tumor cell purification of mouse sarcoma EPO-GEMMs, existing protocols for primary human sarcomas[67] were adapted to mouse specimens. GEMM tumors were thoroughly minced with scissors, taken up in 40 ml of FCS-free DMEM supplemented with 480 µl of 10 mg/ml Trypsin (Sigma, T9935) and 800 µl of 50 mg/ml Collagenase II (Thermo Fisher, 17101015) and subjected to digestion in a water bath at 37 °C for one hour under repeated swirling. If tumor fragments were also desired for allografting, minced tumor fragments were resuspended in FCS-free DMEM and filtered through 200 µm (PluriSelect, 43-50200-50) and 50 µm (PluriSelect, 43-50050-50) cell strainers arranged in a 50 ml falcon tube. This process isolated tumor cell fragments ranging from 50–200 µm, which were then washed into an additional falcon and stored at 4 °C until allografting or freezing. To halt enzymatic digestion, 240 µl of 10 mg/ml Trypsin inhibitor (Sigma, T6522) was added. Next, extracellular DNA was digested by a 1:1 mix of 2 mg/ml DNase (Sigma, D4527) and 1 M magnesium chloride (Fisher Scientific, 15493679) added in 1–4 steps of 60 µl each until viscosity decreased and tumor fragments settled. The solution was filtered through a 40 µm cell strainer (Corning, 352340), centrifuged at $400 \times g$ for 5 minutes, and then resuspended in 2 ml ACK lysis buffer (Thermo Fisher, A1049201) for 2–5 minutes for red blood cell lysis. After two washes with 10 ml of FCS-free DMEM, cells were counted (Countess 3 cell counter, Invitrogen, AMQAF2000) and resuspended for seeding or re-engraftment in suitable media. About 0.5 to $1 \times 10^6$ were seeded in 2 ml of full DMEM (2D culture) or a spheroid TSM-complete for 3D culture[67] into six-well plates (Greiner, 657160 for adherent culture and Thermo Fisher, 174932 for suspension culture). Collagen coating was employed for 2D culture to facilitate attachment for the first passage. After six in vitro passages, cell lines were considered established. Dissociation of cells was performed with Trypsin (Sigma, T4049) for 2D cultures and TrypLE (Invitrogen, 12604-013) for 3D cultures. For cryopreservation, ~1–2 million cells, or 1000–3000 tumor spheroids or fragments, were suspended in Synth-a-Freeze

Cryopreservation Medium (Gibco, A1254201), placed in gradual freezing aid (Thermo Fisher, #5100-0001), and moved to −80 °C before transferal to liquid nitrogen for long-term storage. All in vitro lines were cultured in a humidified incubator at 37 °C with 5% CO2. The MycoAlert Kit (Biozym 883103) was regularly employed to verify the absence of bacterial contamination with Mycoplasma in all cell lines.

### Plasmids and vectors

Constructs cloned for and used in this study are listed in Supplementary Data 4. For most EPO experiments, pSB_EF1a_MCS or pPB_E-F1a_MCS vectors were used to insert oncogenic open reading frames amplified from human cDNA or ordered as gene strands based on publicly available coding sequences. Assembly was achieved by restriction insertion cloning or Gibson assembly using the NEB Hifi kit (E2621L). For luciferase vectors, a point-mutated version reported to be less immunogenic, albeit slightly less bright, was used[68]. Whole-plasmid sequencing was employed to ensure that plasmids maintained the correct sequences. *KRAS* expression was validated in HEK293T cells via Western Blot after transfection with PEI (Thermo Fisher, BMS1003-A). The ASPS model was generated in a collaborative project with Priya Chudasama on "Immunogenomics characterization of alveolar soft part sarcoma".

sgRNAs were ordered as oligonucleotides from Sigma-Aldrich and inserted into the PX330 CRISPR vector (Addgene 42230) by digestion with BbsI. Correct insertion was validated by Sanger Sequencing. sgRNAs were designed using Benchling.com, aiming to target the first common exon across all isoforms with the highest on- and lowest off-target scores. Five guides per target gene were tested in vitro for their editing efficiency in N2A cells after transfection using Lipofectamine 3000 (Thermo Fisher, L3000001). Where available, previously published guides were included. sgRNA sequences are listed in Supplementary Data 5.

For in vivo EPO, vectors were amplified by endotoxin-free Giga prep (Qiagen 12391 or Zymo Research D4204). NEB Stable chemically competent *E. coli* (NEB C3040H) were used and cultured at 32 °C. Giga preps were started from validated glycerol stocks in 6 ml LB cultures, cultured overnight and used to inoculate 2.5 l cultures, harvested the following day for preparation. Final DNA pellets were taken up in about 200 µl of PBS and diluted to ~5–10 µg/µl. For some experiments, the PBS was adjusted to a final concentration of 6 mg/ml of poly-L-glutamate (Sigma-Aldrich, P4761). Plasmids were stored at 4 °C for short-term and −20 °C for long-term storage. For EPO, 4 µg of transposase was mixed with 8 µg of each transposon vector, 8 µg of each CRISPR vector and 8 µg of each reporter vector as outlined in the results. For P30 EPO, the plasmid mixture was adjusted to 1:100 with methylene blue dye (Sigma 50484) in PBS in a final volume of 25 µl per leg. For P0 EPO, methylene blue was used at 1:25 with a final volume of 5 µl. For Sleeping Beauty vectors (SB) SB13 transposase was used as in vitro experiments did not show superiority of SB100X over SB13. For PiggyBac (PB) vectors, PiggyBac transposase was used.

### Western blotting

Cells were harvested, washed with PBS, and resuspended in RIPA buffer (Cell Signaling, 8906) with protease inhibitors (Sigma-Aldrich, 11836170001) for 30 minutes on ice under regular vortexing. After centrifugation (15 minutes at $17,000 \times g$, 4 °C), protein lysates were quantified using the BCA protein assay (Thermo Fisher, 23227). Samples were adjusted, denatured in 2× Laemmli at 95 °C for 5 minutes, and loaded onto 4–15% protein gels (Biorad, Biorad 456-1084) at 30 µg per well. Semi-dry blotting (Biorad, #1704150) transferred proteins to a PVDF membrane, pre-activated with methanol. The membrane was blocked, incubated with primary antibody overnight, washed, and incubated with a secondary antibody. After a final wash, the membrane was incubated with ECL solution (Perkin-Elmer, NEL103001EA) and developed on an Amersham Imager 680. For β-actin antibody (already

HRP-coupled), the secondary antibody step could be omitted. A list of utilized antibodies is delineated in Supplementary Data 6.

## Animals

Animals for this study were purchased from Janvier Laboratories and housed at the central DKFZ animal facility under Specific Opportunist Pathogen-Free (SOPF) conditions, utilizing individually ventilated cages. Animals had *ad libitum* access to food and water. Daily assessments of their well-being were carried out by certified animal caretakers. CD-1 and C57BL6/J mice used for EPO were 4–6 weeks old (P30 condition) or 1–2 days old (P0 condition). Details on utilized mouse strains (CD-1 and C57BL6/J) are given in Supplementary Data 7.

## In vivo EPO

**EPO of P30 (4–6 weeks old) animals.** Preemptive oral analgesia (4 mg/ml Metamizol in drinking water ad libitum, sweetened to achieve a final concentration of 1.5% glucose) commenced 1 day before surgery and persisted for three days thereafter. Additionally, animals received a single dose of 200 mg/kg metamizole subcutaneously under anesthesia. Anesthesia was induced with 1.5–2.5% isoflurane under close monitoring of vital functions before shaving the legs, disinfecting the operating field, and initiating the procedure. A warming mat maintained a stable body temperature. Bepanthen ointment protects eyes from drying or keratitis. Mice were positioned on their backs on a sterile drape with limbs gently secured. The quadriceps femoris muscle was surgically exposed bilaterally. For most experiments, 25 µl of hyaluronidase was intramuscularly injected to enhance transfection efficiency before temporarily closing the skin with wound clips. After a two-hour rest, clips were removed under anesthesia. In experiments without hyaluronidase pre-treatment, muscle exposure was directly followed by EPO. The blue plasmid mixture was injected into the thigh muscle perpendicularly to the muscle fiber direction under visual control using a calibrated Hamilton glass syringe (25 µl, Model 702 RN, CAL7636-01) with a 28 G needle (Removable needle, 28 G, point style 4, 7803-02). Thereafter, the exposed muscle was directly placed between two 5 mm platinum plate electrodes (Nepagene, CUY650P0.5-3), and 5 unidirectional 100 V pulses of 35 ms length with 500 ms intervals (or other conditions as outlined in results) were applied using a calibrated electroporator (Nepagene, NEPA21). The procedure concluded with a final disinfection of the operating field and wound closure through continuous suturing before the animals were placed in a separate cage to recover under close supervision. The procedure was well-tolerated. EPO of P0 (newborn) animals Mother animals received additional nesting material ~1 week before birth to mitigate the risk of offspring rejection post-procedure. Following birth, pups underwent a brief separation of ~10–15 minutes from the dam, during which EPO was conducted under anesthesia with 1.5–2.5% isoflurane. Pups were collectively placed on a warming mat beneath a sterile drape to maintain stable body temperature. For EPO, mice were positioned on their backs, and limbs were gently secured with sterile tape after gentle disinfecting the operation field. Due to the delicate nature of newborn skin, surgical exposure of the muscle was omitted. Instead, 5 µl of plasmid mix, with 1:25 diluted methylene blue, was transcutaneously injected into the thigh muscle of both legs using a calibrated Hamilton glass syringe (5 µl, Model 75 RN, CAL7634-01) with a 30 G needle (Removable needle, point style 4, 7803-07). The higher concentration of methylene blue allowed visually controlled injection, albeit slightly less precise than in P30 mice. Subsequently, the leg was placed between two 5 mm platinum plate electrodes, and 5 unidirectional 70 V pulses of 35 ms length with 500 ms intervals (or other conditions as outlined in results) were applied using the Nepa21 electroporator. Post EPO, mouse pups were reunited with their mother and left undisturbed. The procedure was well-tolerated. All offspring from both CD1 and C57BL6/J mice were consistently well-accepted. EPO cohorts always included at least two genetic groups per litter, and male and female mice were included equally in all analyses. Each genotype combination was tested in at least 6 animals (3 males, 3 females). We also allow at least one week after the mice arrive at our facility before conducting any procedures, further reducing the likelihood that differences in tumor phenotypes are due to batch variations across litters.

## Tumor surveillance in vivo

**Caliper measurements.** In addition to daily health assessments by certified animal caretakers, comprehensive weekly examinations were carried out by scientific staff to detect tumors through thigh palpation and identify associated signs of disease. Regular weight measurements were also performed. Once a palpable tumor emerged, its length and width were consistently measured using a digital caliper (Fine Science Tools, 30087-00), typically three times per week, with daily monitoring for rapidly growing tumors. Tumor volume was determined using the formula: $(\text{length} \times \text{width}^2)/2$. Animals were observed for up to one year post-EPO unless tumor growth reached a maximum of 15 mm in one dimension or other termination criteria were met, which included weight loss of up to 20%, apathy, abnormal posture, piloerection, respiratory problems, and specific signs on the Grimace Scale (constricted eyelids, sunken eyes, flattened ears)[69], invasive growth into thigh muscles causing functional limitations (lameness), disability, or pain, exulcerations or automutilations. Neonatal mice were terminated if they exhibited the absence of a milk spot, cannibalism, rejection by the mother, color change (from pink to blue or pale), or lack of locomotion in response to touch stimuli. Mice were killed by cervical dislocation with or without deep narcosis or increasing $CO_2$ concentrations. Newborns until P5 were killed by decapitation.

**In vivo bioluminescence imaging.** When utilizing luciferase as a reporter gene, regular monitoring of gene expression, transfection efficiency, and tumor growth was conducted using IVIS (In vivo imaging system) bioluminescence imaging. D-luciferin (Enzo, 45784443) was prepared in PBS (15 mg/ml), sterile-filtered, aliquoted into light-protected vials, and stored at −20 °C until use. Mice were anesthetized with 1.5–2.5% isoflurane, injected intraperitoneally (i.p.) with luciferin (10 µl/g of mouse weight), and positioned in the IVIS chamber on their backs with isoflurane administered through a mouthpiece. Bepanthen ointment was used to protect their eyes. A 10-minute incubation period was followed by 7 minutes of image acquisition, determined to be optimal during the plateau phase of the IVIS signal based on pilot experiments. Typically, three animals were imaged simultaneously. When four to five animals were imaged concurrently, an XFOV-24 lens was used. After imaging, mice were placed in a separate cage to recover. Living Image software (PerkinElmer, version 4.5.5) was used for analysis. Regions of interest (ROIs) were defined around the electroporated regions to quantify IVIS signal intensity as photons/second.

## Syngeneic allografting of mouse tumors

Primary and in vitro cultured murine tumor material from EPO-GEMMs was purified and dissociated as outlined above (cell culture section), quantified and syngeneically re-engrafted into wild-type C57BL/6 J recipient female mice aged 4–8 weeks. Dissociated cells were counted using an automated cell counter (Countess 3, Invitrogen, AMQAF2000), while 50–200 µm, sized and tumor fragments and spheroids were manually counted in $10 \times 5$ µl droplets on a petri dish under a light microscope (Zeiss, 491237-9880-010). The mean count of five droplets was used to estimate the total number of fragments or spheroids. Cells, spheroids, and fragments were then centrifuged and resuspended at concentrations of $1 \times 10^6$ primary or cultured cells and 1000 spheroids or fragments per 12.5 µl of FCS-free DMEM, which were kept on ice until injection. Perioperative preparation was performed analogously to P30 EPO, except for shaving, which was exclusively performed on the left leg. Mice were placed on their backs, and their

extremities were gently fixed with sterile tape. A sterile pen was positioned diagonally beneath the left leg to expose the thigh muscle. Just before engraftment, solutions containing tumor fragments, tumor cells, or tumor spheroids were resuspended ice in 1:1 ratio with Matrigel (Corning, 354277 or Thermo Fisher, A1413202) and loaded into a calibrated Hamilton glass syringe (25 μl, Model 702 RN, CAL7636-01) equipped with a 28 G needle (Removable needle, 28 G, point style 4, #7803-02). The injection was performed transcutaneously into the thigh muscle perpendicularly to the muscle fiber direction under visual control. Subsequently, mice were placed in a separate cage to recover.

## Preclinical treatment studies

**NTRK inhibitor treatment in vitro.** 500–1000 cells per well, cultured in full DMEM, were seeded into black flat-bottom 384-well plates (Greiner, 781091) for each cell line based on preliminary experiments to determine the appropriate cell numbers based on baseline growth rates. Each drug was pre-printed into 384-well plates in 10 doses, ranging from 0 to 10,000 nM of Larotrectinib (Medchem Express, HY-12866) and Repotrectinib (Medchem Express, HY-103022), taken up in 99.9% DMSO (Sigma, D8418) in semi-logarithmic increments (in amounts of 0, 1, 3.16, 10, 31.6, 100, 316, 1000, 3160, 10000 nM). 100 μM of Benzethonium chloride (Sigma, PHR1425) was used as a positive control. Staurosporin (TargetMol, T6680) as a dose-dependent therapy response control with one technical replicate per dose and cell line (in amounts of 0.1, 1, 10, 100, 1000 nM). For all other groups, four technical replicates were used per dose and drug for each cell line. The edges of the plates were filled with PBS to avoid edge effects. Order of drugs and doses was based on a random distribution to avoid batch effects. Treatment effects were quantified 72 hours after cell seeding using calorimetric ATP measurement with CellTiterGlo 2.0 (Promega, G7572) according to the manufacturer's protocol and as described previously[67] using an Infinite M Plex plate reader (Tecan). Data was analyzed using the previously described iTrex algorithm[70] for quality control and to condense dose-response curves into representative drug sensitivity scores (DSS) ranging from 0 to 50, 0 indicating resistance, >10 indicating sensitivity.NTRK-inhibitor treatment in vivoTumors were induced by engrafting 50,000 murine tumor cells from a FD-IFS mouse tumor-derived cell line (2D) into the left thigh muscle of 7-week-old female C57BL/6 J mice. Treatment was started 8–10 days after tumor cell transplantation when all mice exhibited palpable tumors, and continued for a maximum of three weeks with oral gavage of larotrectinib, repotrectinib or vehicle alone (0.5% carboxymethylcellulose and 1% Tween-80 in water) twice per day. 10 μl/g body weight of vehicle or 2 mg/ml inhibitor-vehicle solution was administered. Tumor-bearing mice were randomized for treatment.

## Tissue removal and processing

Euthanized mice were carefully examined for metastatic disease of internal organs after removing the skin. Tumors were photographed. Some animals underwent macroscopic examination of GFP expression using a fluorescent dissection microscope (Leica, 25716). Tumors were surgically separated from the femur and surrounding organs. To prevent cross-contamination between tumors, all instruments were meticulously disinfected before handling a new tumor. The tumor was dissected into multiple sections on an inverted Petri dish using a microtome blade (VWR, 720-2369). One cross-section was placed in a histo cassette (Sigma, H0792-1CS) and fixed in formalin for 2–3 days before transfer to 50% (v/v) ethanol at 4 °C until further processing. Another cross-section was embedded in cryo-embedding resin (OCT compound, Tissue-Tek, 14291) and placed in a cryomold (Tissue-Tek, 14292), which was set on a metal rack over dry ice for homogeneous snap freezing before transfer to −80 °C. The remaining tissue was sectioned into small tumor pieces (20–30 mg), with portions allocated

for nucleic acid purification for molecular analysis (snap-frozen on dry ice) and tumor cell isolation for cell culture and allografting into recipient mice.

## Nucleic acid extraction

Before nucleic acid extraction, all surfaces and instruments underwent cleaning with RNase decontamination reagent (Thermo Fisher, 7000TS1). DNA and RNA were extracted from the same tumor tissue piece, ~20–30 mg in weight, using Qiagen's DNA/RNA AllPrep kit (80204) following the manufacturer's protocol. If only DNA was required, the Qiagen DNeasy kit (69506) was employed for purification. Tissue homogenization was accomplished using DNAse/RNAse-free pestles (Carl Roth, CXH7.1), swirled in 1.5 ml Eppendorf tubes with a cordless pestle motor (DWK Life Sciences, 749540-0000). Lysates were subsequently passed through a QIAshredder column (Qiagen, 79654) to eliminate debris before proceeding with nucleic acid purification. Purified DNA and RNA were maintained on ice, concentration was determined using Nanodrop and Qubit, and then stored at −20 °C (DNA) and −80 °C (RNA) for subsequent analyses.

## Oncogene PCR from genomic DNA

Genomic DNA, extracted from tumor samples and plasmid DNA as positive controls, served as the template for polymerase chain reaction (PCR). 1 μl of genomic or plasmid DNA, containing 10–100 ng of DNA, was combined with 12.5 μl of 2× Red HS Mastermix (Biozym, #331126 L), 1.25 μl of the forward primer, 1.25 μl of the reverse primer, and 9 μl of water. PCR products were loaded onto 0.7-1% (w/v) agarose gels, along with 1 kb (NEB, N3232L) and 100 bp (NEB, N3231) DNA ladders. Electrophoresis was conducted at 80–110 V until satisfactory separation was achieved, followed by imaging using a Gel doc imager (Biorad, 170-8170). Primer sequences and PCR conditions for genotyping PCRs are outlined in Supplementary Data 8.

## Indel analysis from genomic DNA

To assess the efficacy of CRISPR-mediated editing of tumor suppressor genes, primers were designed to amplify the corresponding genomic loci spanning ~500–700 base pairs around the sgRNA target site. PCR products were purified using NucleoSpin Gel and PCR Clean-up Kit (MACHEREY-NAGEL, 740.609.250) and subsequently subjected to Sanger sequencing through Microsynth Seqlab services, utilizing the corresponding forward primer. The obtained sequences were aligned with the wildtype sequences derived from mouse tail genomic DNA, using the TIDE algorithm (Tracking of Indels by Decomposition)[71] (http://shinyapps.datacurators.nl/tide/) to calculate the percentage of insertions and deletions. Primer sequences and PCR conditions for TIDE PCRs are outlined in Supplementary Data 5.

## Immunohistochemistry and multiplexed immunofluorescence imaging

**Immunohistochemistry.** After fixation in 10% (v/v) buffered formalin (Sigma, HT501128) for 2–3 days, tissue sections underwent a stepwise rehydration process in an alcohol series using an automated tissue processor (Leica, ASP300S) until reaching a concentration of ≥99% (v/v) ethanol. Subsequently, the tissue cassettes were transferred to intermediate xylene and embedded in paraffin (Leica, 14039357258). In all, 3–4 μm paraffin sections were prepared with a HM 355S microtome (Fisher Scientific, 10862110), deparaffinized and rehydrated up to 96% ethanol (v/v). H&E and PAS stainings were performed with standard protocols. For H&E, 5 mins of Haemalaun nuclear staining (Carl Roth, T865.3) was followed by 5 mins of rinsing with water and 20–30 s of Eosin solution (Merck, 115935, 100 ml supplemented with one drop of acetic acid, Merck, 1.00063). For PAS staining, periodic acid (Merck, 100524) was applied for 10 mins, Schiff's reagent (Merck, 1.09033) for 5 mins, Haemalaun nuclear staining (Carl Roth, T865.3) for 1 minute, and 5 mins of rinsing. For

immunohistochemistry, primary antibodies were typically diluted in antibody Diluent (DAKO, S2022) with 2% milk (v/v) for 90 mins at 37 °C, followed by four washes with TBST-T and application of bio-tinylated secondary antibodies against the species of the primary antibody, typically diluted 1:500 in TBS-T for 30 mins at 37 °C. This was followed by three washes and treatment with 1:200 alkaline phosphatase streptavidin (Vector, SA-5100) in TBS-T for 30 mins at 37 °C. Signal amplification, if necessary, involved another round of secondary antibody and alkaline phosphatase streptavidin treatment. Finally, substrate red (Dako, K5005) was applied for 10 mins at room temperature before slides were washed, counterstained with Haemalaun (Carl Roth, T865.3), and mounted with Aquatex (Sigma, 1.08562.0050). Antibodies and corresponding antigen retrieval methods are listed in Supplementary Data 6. Slides were scanned using the Aperio AT2 slide scanner (Leica, 23AT2100) at ×40 magnification. Morphological assessment and quantification of immunohistochemistry and morphological tumor features were conducted with the expertise of pathologist Felix Kommoss in a blinded fashion. Digitized image files were analyzed using QuPath software (version 0.4.1) and assembled into final figures using Affinity Designer software (version 1.10.5).Multiplexed immunofluorescence imagingCryo-embedded mouse tumors (OCT compound, Tissue-Tek, #14291) were cut to 4–5 μm with a cryostat (Leica CM 1950, Leica Biosystems) and positioned onto SuperFrost plus slides (R.Langenbrinck, 03-0060). Slides were stored at −80 °C until use. Tissue fixation was performed with 4% paraformaldehyde (Thermo Scientific, J19943.K2) for 10 minutes at room temperature. Subsequently, slides were mounted onto the respective MACSwell imaging frame (Miltenyi Biotec, 130-124-673), washed 3 times with MACSima™ Running Buffer (Miltenyi Biotec, 130-121-565) before nuclear staining was performed with DAPI (Miltenyi Biotec, 130-111-570) diluted 1:10 in running buffer. Slides were washed three times with running buffer and submitted to MICS (MACSima™ imaging cyclic staining). Antibodies were diluted with Running Buffer in a MACSwell™ Deepwell Plates (Miltenyi Biotec, 130-126-865) in a total volume of 1050 μl. Spatial analysis using MICS comprised fully automated iterative cycles of ultra-high content immuno-fluorescence based on fluorochrome-labeled antibody staining, image acquisition, and fluorochrome removal. Images were generated according to the manufacturer's instructions and as described before[72]. In brief, ROI was defined based on the DAPI signal and the focus was set using hardware autofocus settings. Raw data was processed with the MACS® iQ View image analysis software (Version 1.2.2) as described before[72]. Processing included automated optimal exposure time selection, calibration correction, stitching of fields of view (FoVs), and subtraction of pre-stain bleach images. Image quality control was assessed, data was analyzed, and MICS data was visualized using MACS® iQ. Utilized antibodies and dilutions are detailed in Supplementary Data 6.

### DNA methylation and CNV analysis of murine tumors and tissue
Genome-wide methylation profiling was performed using the Illumina Infinium Mouse Methylation BeadChip covering >285,000 CpG sites distributed over the mouse genome. A minimum of 500 ng of genomic DNA extracted from frozen tissue was submitted per sample. DNA quality was ensured by digital gel electrophoresis (Agilent, G2939BA). IDAT files were obtained and processed using the R package sesame, version 1.16.1 (SEnsible Step-wise Analysis of DNA MEthylation BeadChips)[73] to generate normalized beta values. Probe intensities underwent background correction using the $p$ value with the out-of-band array hybridization approach, followed by a normal-exponential out-of-band approach. Dye bias correction was performed by aligning green and red to the midpoint using the dyeBiasCorrTypeINorm method in sesame. Probes targeting the X and Y chromosomes were excluded. Clustering analysis was performed with R packages Rtsne

and umap based on the 2000 or 10,000 most variably methylated probes, with perplexity values set to 5 for tSNE and 15 for umap clustering. To infer CNV profiles, the method described in R package conumee, version 1.32.0, was adapted for the mouse array. Specifically, a panel of $n = 60$ normal tissue idat files from C57BL6 mice, kindly provided by Marc Zuckermann and Tuyu Zheng (DKFZ) underwent the same sesame correction pipeline, and total probe intensities were quantified across all probes in tumor and normal samples. The background ratio of cancer sample to normal control intensities was determined using the slope of a linear model. Subsequently, the log base 2 ratio of observed vs. expected intensity was calculated for every probe. Probes were binned using sesame according to their mm285 array manifest, utilizing the getBinCoordinates function. For heatmap visualization of CNV, the color range representing the log fold change in probe intensities was set to −1 to +1.

### RNA sequencing analysis of murine tumors and tissue
RNA exclusively sourced from fresh frozen tissues was quality-controlled by digital gel electrophoresis (Agilent, G2939BA). Only samples with RNA integrity scores (RIN) exceeding 7 proceeded to library preparation using the TruSeq Stranded total mRNA protocol, starting with 50 μl of undegraded RNA at concentrations ranging from 50 to 80 ng/μl per sample. Sequencing was carried out on Illumina's NovaSeq 6000 S1 or S4 flow cell with paired-end 50 bp reads, yielding an average of ~20 million reads per sample. Sequencing reads were aligned to the mouse reference genome (GRCm38mm10) by DKFZ's Omics IT and Data Management Core Facility using the One Touch Pipeline (OTP)264/RNAseq workflow pipeline, version 1.3.0 (STAR Version 2.5.3a, Merging/duplication marking program: Sambamba Version 0.6.5. SAMtools program: Version 1.6.), resulting in raw counts, RPKM, and TPM values. K-means analysis was performed using the iDEP web collection of R packages[74]. Further differential gene expression analysis was conducted with DeSeq2 in R Studio, version 2022.07.1. GO analysis for genes in k-means cluster was performed using DAVID. For cell deconvolution, RNA Seq data was processed using CibersortX in accordance with the developer's manual (https://cibersortx.stanford.edu)[75]. The 'Tabula muris' dataset, a publicly accessible collection of single-cell RNA sequencing data from mice, including >100,000 annotated single cells representing more than 130 cell types across 20 different organs[76] was used to create a reference matrix for cell type annotation. Batch correction parameters were set to S mode (recommended for single-cell reference data).

### Cross-species transcriptome analysis of mouse EPO-GEMMs and human sarcomas
To compare RNA-sequencing profiles between mouse and human sarcoma specimens, data of human sarcomas (TCGA, St. Jude and INFORM), representing common and rare soft-tissue sarcoma entities, was assessed for the top 500 most differentially expressed genes in DEseq2 v1.30.1 (log$_2$FC > 2, FDR < 0.05) in each subtype vs all other subtypes. The top 500 most differentially expressed genes were determined in the same way for $n = 63$ mouse samples of primary GEMMs representing 10 subtype entities. All selected human genes were matched to orthologs in mouse genes. In cases with no orthologs, these genes could not be considered in the comparative analysis. The lists of mouse genes and orthologs from human genes were combined to a total of 2636 genes (Supplementary Fig. 8a, b). Alternatively, all ortholog genes identified and the top 2000 most variable genes were selected (Fig. 5). These gene lists were subsequently used to create matrices and batch correction was performed for species, sample type ("Muscle control") and dataset (St. Jude, TCGA and INFORM) using Harmony v0.10[77], implemented in R v4.0.3. These batch-corrected values were then subjected to t-SNE clustering, excluding non-relevant entities that could not be clearly assigned to one entity or did not have obvious matching mouse samples (e.g., Osteosarcoma, EwS).

To compare methylome profiles between mouse and human specimens, data from 302 human sarcomas from the DKFZ Sarcoma methylation classifier were used, representing 12 distinct sarcoma entities. Comparison of syntenic probes was undertaken, similar to prior analysis described by Zhou et al.[48]. UCSC liftOver was used to map the Infinium HumanMethylation450 array probe sites from the hg19 to mm10 reference genome. There were 15218 shared syntenic probes identified as those overlapping with MM285k array sites. Beta values from these syntenic probes were computed across human methylomes from the DKFZ Sarcoma classifier and the methylomes of EPO-GEMMs. These were converted to M values. Probes were selected for those significantly associated with at least one tumor subtype ($p < 0.001$) and anti-correlated to species ($R^2 > 0.8$). Harmony was used to project all samples onto a species-neutral space by correcting for the species as a batching variable. The resulting corrected M value matrix was used to generate the reported t-SNE figures.

## Statistical analysis

Statistical analysis was performed as outlined in the respective figure captions for each experiment using R v4.0.3. Wherever suitable, the Bonferroni–Holm method was used to correct for multiple testing. The provided sample size ($n$) indicates biological replicates. Group sizes for in vivo experiments were determined through statistical consultation in the Department of Biostatistics at the DKFZ, which included in silico simulations. The H&E histographs are representative of at least $n \geq 3$ tumors that were considered for comparison by expert pathologist review, depending on the number of tumors that developed per tumor type. For most subtypes, $n \geq 5$ tumors were assessed. Group allocation for preclinical treatment studies was determined through random distribution. Outcome assessment for preclinical treatment trials was performed in a blinded fashion. Other group allocations and outcome assessments were performed in a non-blinded fashion. Survival was measured using the Kaplan–Meier method and log-rank tests. The threshold for significance was set to $P < 0.05$.

## Figure preparation

Data was plotted using R v4.0.3 and Graphpad Prism version 8.4.3. Affinity Designer version 1.10.5. and Biorender.com were used to generate graphical illustrations and arrange panels into final figures.

## Reporting summary

Further information on research design is available in the Nature Portfolio Reporting Summary linked to this article.

## Data availability

The data generated in this study have been deposited at the Gene expression omnibus (GEO) under accession number GSE265875. The allografts models described in this study are available for preclinical testing through the Innovative Therapies for Children with Cancer Pediatric Preclinical Proof-of-concept Platform (ITCC-P4) (https://itccp4.com/). Source data are provided with this paper.

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

## Acknowledgements

We thank Daisuke Kawauchi, Joseph Leibold and Kerstin Dell for insightful discussions regarding the development of a muscle-specific EPO method and members of the soft-tissue sarcoma lab for feedback and fruitful discussions. Darjus Tscharganeh, Hsuan-An Chen (Scott Lowe lab) and Rienk Offringa for sharing reagents. We thank the light microscopy core facility, the genomics and proteomics core facility and the small animal imaging center, along with the outstanding and dedicated team of animal caretakers at the DKFZ, for their exceptional support and services. We thank Tim Holland-Letz and the Department of Biostatistics of the DKFZ for statistical consultation and Norman Mack (Pfister Laboratory) for his support with various animal protocols. We thank Veronika Eckel and Alexander Brobeil from the Tissue Bank of the National Center for Tumor Diseases (NCT), Heidelberg, Germany, for their excellent services in scanning histology slides. We express our gratitude to Marie Koloseus for her preliminary work in establishing a mouse antibody panel for multiplexed immunofluorescence imaging. Veronika Bahlinger, Christian Schürch and Benjamin Mayer for supplying additional human histology specimens and Katharina Hartwig for her valuable contribution to graphic design. We thank all INFORM study group leaders and national coordinators. This project has received funding from the European Research Council (ERC) under the European Union's Horizon 2020 research and innovation program (grant agreement n° 805338) (A.B.) as well as the Physician-Scientist Program of the University of Heidelberg, Faculty of Medicine (R.I.). D.B. is supported by the DKFZ International PhD Program. Work in the laboratories of A.B., T.G.P.G., S.M.F., P.C. and C.S. is supported by the HEROES-AYA (Heterogeneity, Evolution, and Resistance in Oncogenic Fusion Gene-Expressing Sarcomas Affecting Adolescents and Young Adults) consortium within the National Decade Against Cancer of the German Federal Ministry of Education and Research (01KD2207). T.G.P.G. acknowledges support from the Barbara and Wilfried Mohr foundation. P.C. receives support from an Emmy Noether Program Grant CH2302 of the DFG (German Research Foundation). Finally, we express our deepest gratitude to all pediatric and adult patients affected by sarcomas who have provided us with inspiration and motivation to pursue this work.

## Author contributions

R.I. conceived the study, designed, performed, and analyzed the experiments, and wrote the manuscript. D.B., S.P., M.H., S.S., C.S., F.H.G., M.v.E., and F.C-A. performed and analyzed experiments. F.K.F.K., T.G.P.G., and C.V. performed expert histopathology review. E.S.Z., M.S., and C.B. performed bioinformatic analyses. D.L., L.S., L.W., C.W., and C.Schm. assisted with experiments. H.W., J.F., and S.L. assisted with preclinical studies. I.Oe., H.P., K.B.J., P.Ch., C. Scho., and P.G. provided essential materials and information. S.M.P. acquired resources, funding and did a writing review. R.A. supervised and conducted bioinformatic analyses and edited the manuscript. A.B. conceived and coordinated the study, designed experiments, acquired funding and wrote the manuscript. All authors read, revised and approved the final manuscript for publication.

## Funding

## Competing interests

I.O. receives research grants from PreComb, BVD and Day One Therapeutics. The remaining authors declare no potential conflict of interest.

## Additional information

¹Soft-tissue sarcoma research group, German Cancer Research Center (DKFZ), Heidelberg, Germany. ²Hopp Children's Cancer Center Heidelberg (KiTZ), Heidelberg, Germany. ³National Center for Tumor Diseases (NCT), NCT Heidelberg, a partnership between DKFZ and Heidelberg University Hospital, Heidelberg, Germany. ⁴Department of Pediatric Oncology, Hematology and Immunology, Heidelberg University Hospital, Heidelberg, Germany. ⁵Division of Pediatric Surgery, Department of General, Visceral and Transplantation Surgery, University Hospital Heidelberg, Heidelberg, Germany. ⁶Faculty of Biosciences, University of Heidelberg, Heidelberg, Germany. ⁷Institute of Pathology, University of Heidelberg, Heidelberg, Germany. ⁸Division of Pediatric Neurooncology, German Cancer Research Center (DKFZ), Heidelberg, Germany. ⁹German Cancer Consortium (DKTK), DKFZ, core center Heidelberg,

Heidelberg, Germany. [10]Department of Neuropathology, University Hospital Heidelberg, Heidelberg, Germany. [11]Clinical Cooperation Unit Neuropathology, German Cancer Research Center (DKFZ), Heidelberg, Germany. [12]Core Facility Tumor Models, German Cancer Research Center (DKFZ), Heidelberg, Germany. [13]Division of Vascular Oncology and Metastasis, German Cancer Research Center (DKFZ-ZMBH Alliance), Heidelberg, Germany. [14]Clinical Cooperation Unit Pediatric Oncology, German Cancer Research Center (DKFZ), Heidelberg, Germany. [15]Department of Pediatric Hematology and Oncology, University Hospital Tuebingen, Tuebingen, Germany. [16]Division of Translational Pediatric Sarcoma Research, German Cancer Research Center (DKFZ), Heidelberg, Germany. [17]Faculty of Medicine, Heidelberg University, Heidelberg, Germany. [18]Section of Pediatric Pathology, Department of Pathology, University of Bonn, Bonn, Germany. [19]Precision Sarcoma Research Group, German Cancer Research Center (DKFZ), Heidelberg, Germany. [20]Division of Applied Functional Genomics, German Cancer Research Center (DKFZ), Heidelberg, Germany. [21]Core Facility Light Microscopy, German Cancer Research Center (DKFZ), Heidelberg, Germany. [22]Department of Orthopaedics, University of Utah School of Medicine, Salt Lake City, UT, USA. [23]Department of Oncological Sciences, Huntsman Cancer Institute, University of Utah School of Medicine, Salt Lake City, UT, USA. ✉e-mail: robert.autry@kitz-heidelberg.de; a.banito@kitz-heidelberg.de

