## [Transparent Peer Review file · Nature Communications]

Somatic gene delivery faithfully recapitulates a molecular spectrum of high-risk sarcomas

Corresponding Author: Dr Ana Banito

Version 0:

Reviewer comments:

Reviewer #1

(Remarks to the Author)

I thank the authors for responding to my comments and suggestions. Overall, reframing the analysis based on the genetic alterations within each model rather than the human sarcomas they are supposed to represent has helped to alleviate many of my concerns about the manuscript. The adjustments and clarifications on the bioinformatic analysis have also substantially strengthened the authors' conclusions. Without additional data, I would favor removing the CAR-T experiments. Although the authors present limited new insights into sarcoma biology at present, the optimized in vivo electroporation protocol and many EPO-GEMM models generated here will be a helpful resource for the sarcoma community. Of note, the authors should detail in the manuscript that the models are available via the ITCCP4 consortium and provide information about how they can be requested.

Reviewer #2

(Remarks to the Author)

The current version of the Imle et al. paper remains largely similar to the previous one, with a reduced emphasis on the immune aspects of the tumor microenvironment—despite this being the primary advantage of developing syngeneic mouse models. If the authors do not intend to focus on characterizing the immune system, these models primarily serve to assess how specific treatments behave in the presence of an intact immune system, compared to immune-compromised models such as PDX. However, the paper lacks a direct comparison between these two settings, making it difficult to demonstrate any meaningful differences or similarities.

That said, these models still offer advantages, including scalability, ease of generation, and applicability to certain sarcoma subtypes, especially pediatric sarcoma, for which alternatives may not be available. While I believe these models may have the potential to uncover previously overlooked vulnerabilities in sarcomas, the authors have not fully highlighted their key novelties, such as the ability to provide insights into specific immune tumor microenvironment characteristics that are otherwise unachievable with most conventional mouse models.

In summary, the study is well-executed, and the paper presents high-quality data. However, the broad characterization of multiple mouse models lacks a clear direction on how these models will be crucial in addressing unmet clinical needs.

Reviewer #3

(Remarks to the Author)

The authors have attempted to address initial critiques as to the faithfulness of the mouse models in recapitulating human sarcomas by removing the descriptive naming of each model (MPNST, UPS, etc.) and now describing each model by the specific gene alterations introduced. In the revised manuscript, the authors do acknowledge the imperfect nature of these tumors in imitating human disease and suggest they may have value for modeling the contributions of different tumor suppressors such as Bcor to tumorigenesis. I appreciate these revisions, though I do find that with all these different stories, the manuscript feels a bit dis cohesive and muddled. I think it would be easier to follow from a text perspective (if challenging from a figure organization perspective) to present the full story about the fusion-associated tumors including their

microenvironment, transcriptomic and methylomic similarity to human sarcomas all together before moving on to the less convincing models to discuss their histology, biology and potential uses.

Given that the tumors are now renamed by genotype rather than aspirational histology, the question remains: do these models in fact recapitulate any human sarcomas if they do not recapitulate the tumors they were previously named after, and if not, how can they help to understand human sarcomagenesis or serve as preclinical models for therapy?

I will restrict my comments to the models I took issue with in the original review

KRAS + p53. (now no longer described as ERMS). Despite renaming the tumor model, the manuscript continues to imply that these are recapitulating anaplastic ERMS (as in the following sentence), "KRAS/sgTrp53 tumors, exhibited focal or diffuse anaplasia often observed in patient specimens of TP53-mutated RMS." As noted in the revised manuscript and in the Reviewer's Figure 2, these do not well recapitulate ERMS (regardless of anaplastic features) morphologically or any other sarcoma given negative desmin, myogenin, and nonspecific clustering with human vaguely myogenic tumors. Human sarcomas aside from ERMS rarely have KRAS mutations, although RAS pathway dysregulation may be seen by other mechanisms (for instance in MPNST, IFS, various kinase-altered sarcomas). While KRAS mutations are well known to cause sarcomas in mice, I have always regarded the relevance to human tumors to be dubious. This study does little to assuage this concern. Labeling the tumor only by its genotype does not make it more relevant to human disease, nor are KRAS mutant mouse models of sarcoma novel.

My original critique of the PAX3::FOXO1 model was that it lacks strong/diffuse myogenin staining, a feature of human ARMS. The authors' rebuttal notes that other published mouse models of PAX3::FOXO1 RMS also only show focal myogenin/myod1 and that their model clusters with ARMS in the transcriptomic analysis, and expressed high mRNA levels of myod1 and myog and exhibit high levels of FGFR4. In this instance, I will accept that this model may have value, but if other mouse models of ARMS exist, how does this one improve or add value to those?

KRAS – empty vector ERMS: The case depicted in 3b could be any undifferentiated spindle cell sarcoma. As mentioned above, I am unconvinced that KRAS mutant models of sarcoma in mouse are a good model for human disease, and they are definitely not novel (albeit the existing ones are engineered through other means than electroporation).

KRASG12V/Bcor. Again, the authors have dealt with nomenclature concerns by removing the descriptive label of ERMS. The question still remains, what model of human sarcoma is this tumor useful for? The original aim was to generate a model of the minor subset of ERMS with BCOR mutations, but this model is not that, and it does not seem to recapitulate any known human sarcoma type. This model may be novel, but I am not entirely convinced this will help to better understand human sarcomas, in its present form. The authors suggest that the value of the model is to show that bcor inactivation cooperates with KRAS in tumorigenesis, and that bcor inactivation allows oncogenesis in different cells, resulting in a different methylation pattern. I would think it just as likely that bcor inactivation results in epigenetic reprogramming, as the authors have not put forth clear evidence as to what specific electroporated cells are giving rise to any of these models.

The histology of NTRK-pleo tumors is nonspecific and could be any high grade undifferentiated sarcomas. By mRNA profiling, it clusters with other sgP53 tumors with a pleomorphic morphology (Reviewer's Figure 2). However, none of these sgp53 tumor models clusters with human sarcomas most likely to show p53 mutations/loss (UPS, LMS, MFS, MPNST). As a result I am not sure what to make of these tumor models. The revised text however, does acknowledge that the cell of origin may be deterministic in these mouse models, affecting how well they are able to recapitulate human tumors.

sgNF1/sgtrp53: The renamed model could potentially be a model for some types of human RMS, which do sometimes have NF1 loss, so I will accept this model could have value.

Additional minor critiques

Line 267 "synchronous Smarcb1 inactivation, pathognomonic for Malignant Rhabdoid Tumors (MRT) and Epithelioid Sarcomas (EpS)," SMARCB1 activation, while present in MRT and epithelioid sarcomas, is widely seen in a variety of other malignancies, including carcinomas, a subset of myoepithelial tumors, epithelioid schwannomas and epithelioid MPNST, an unusual vulvar malignancy with yolk-sac-tumor-like differentiation, and many others. Smarcb1 deficiency may well accelerate tumor development in this model, but the phenotype of the tumor it is driving is unclear. I would probably tone down the text to say that the tumors with Smarcb1 inactivation share the epithelioid to rhabdoid histology typical of the various human tumors with this alteration, although the exact histotype seen here is unclear (as it honestly also is in many human SMARCB1 deficient tumors).

I note that while the tumor models have been renamed as per their genetic features, the figures, (for example figure 2 b/d) very clearly attempt to draw parallels to human tumors with the same alterations (e.g. ERMS) which these lesions do not in fact recapitulate). I feel that this is somewhat misleading to the reader.

Lines 556-558 "Most importantly, the somatically induced sarcoma models were faithful representatives of previously established conventional GEMMs and the human disease. They reproduced entity-specific histologies with remarkable accuracy and activated expression programs that reflect the biology of the human disease" is only true for some of the models – the way this is phrased implies that all the models are accurate representations of human disease, which the authors have already acknowledged is not the

case for the mutation-associated models described above. This language should be toned down or made more specific to the fusion models such as SS18::SSX, ASPCR1::TFE3.

Overall, I do believe that this resource could be valuable. However, my suggestions would be to simplify the claims and descriptions of the various *kras/bcor/nf1/p53* / *nrk-pleo* models to make the manuscript easier to follow. I think it would be reasonable to say that while the tumors showed imperfect correlation to specific human tumors, they may be used as a resource to better understand the role of these mutations in transforming precursor cells.

RESPONSE TO REVIEWERS' COMMENTS

Reviewer #1 (Remarks to the Author):

I thank the authors for responding to my comments and suggestions. Overall, reframing the analysis based on the genetic alterations within each model rather than the human sarcomas they are supposed to represent has helped to alleviate many of my concerns about the manuscript. The adjustments and clarifications on the bioinformatic analysis have also substantially strengthened the authors' conclusions. Without additional data, I would favor removing the CAR-T experiments. Although the authors present limited new insights into sarcoma biology at present, the optimized in vivo electroporation protocol and many EPO-GEMM models generated here will be a helpful resource for the sarcoma community. Of note, the authors should detail in the manuscript that the models are available via the ITCCP4 consortium and provide information about how they can be requested.

Thank you for your comments. We removed the CAR-T experiments as suggested. We have added a sentence to the data availability section: "The allografts models described in this study are available for preclinical testing through the Innovative Therapies for Children with Cancer Paediatric Preclinical Proof-of-concept Platform (ITCC-P4) (<https://itccp4.com/>)."

Reviewer #2 (Remarks to the Author):

The current version of the Imle et al. paper remains largely similar to the previous one, with a reduced emphasis on the immune aspects of the tumor microenvironment—despite this being the primary advantage of developing syngeneic mouse models. If the authors do not intend to focus on characterizing the immune system, these models primarily serve to assess how specific treatments behave in the presence of an intact immune system, compared to immune-compromised models such as PDX. However, the paper lacks a direct comparison between these two settings, making it difficult to demonstrate any meaningful differences or similarities.

That said, these models still offer advantages, including scalability, ease of generation, and applicability to certain sarcoma subtypes, especially pediatric sarcoma, for which alternatives may not be available. While I believe these models may have the potential to uncover previously overlooked vulnerabilities in sarcomas, the authors have not fully highlighted their key novelties, such as the ability to provide insights into specific immune tumor microenvironment characteristics that are otherwise unachievable with most conventional mouse models.

In summary, the study is well-executed, and the paper presents high-quality data. However, the broad characterization of multiple mouse models lacks a clear direction on how these models will be crucial in addressing unmet clinical needs.

Thank you for your comments. We understand and appreciate the reviewer's point regarding the importance of characterizing the immune microenvironment across different models. The characterization of the models includes data on overall immune infiltration (CD45 staining) and immune deconvolution from bulk RNA sequencing. As noted in our previous response,

additional characterization using single-cell analysis is currently underway; however, this approach requires significant time and resources. We also don't think that these immunocompetent models should replace PDXs models, nor are they better. Each will have its advantages and disadvantages. The EPO-GEMMs will be particularly useful when it comes to testing immunotherapies or a particular treatment that relies on an immune response. Additionally, given the comprehensive characterizations already presented, the manuscript is quite detailed and extensive. Finally, and as the reviewer noted, the advantages of the EPO-GEMM approach in modeling sarcoma extend beyond its utility for testing treatments, making it valuable for researchers studying sarcoma biology more broadly.

Reviewer #3 (Remarks to the Author):

The authors have attempted to address initial critiques as to the faithfulness of the mouse models in recapitulating human sarcomas by removing the descriptive naming of each model (MPNST, UPS, etc.) and now describing each model by the specific gene alterations introduced. In the revised manuscript, the authors do acknowledge the imperfect nature of these tumors in imitating human disease and suggest they may have value for modeling the contributions of different tumor suppressors such as Bcor to tumorigenesis. I appreciate these revisions, though I do find that with all these different stories, the manuscript feels a bit discohesive and muddled. I think it would be easier to follow from a text perspective (if challenging from a figure organization perspective) to present the full story about the fusion-associated tumors including their microenvironment, transcriptomic and methylomic similarity to human sarcomas all together before moving on to the less convincing models to discuss their histology, biology and potential uses.

Thank you for your comments and suggestions. We have carefully considered your suggestion; however, discussing and comparing the models in terms of their microenvironment, transcriptomics, and methylome for fusion-driven and other sarcomas separately would significantly lengthen and complicate the manuscript. Additionally, we do not believe the models can be categorized as simply convincing or less convincing. Those with more complex genetics, which are inherently more molecularly heterogeneous, are more challenging to model compared to fusion-driven sarcomas, which are molecularly more defined.

Given that the tumors are now renamed by genotype rather than aspirational histology, the question remains: do these models in fact recapitulate any human sarcomas if they do not recapitulate the tumors they were previously named after, and if not, how can they help to understand human sarcomagenesis or serve as preclinical models for therapy?

Yes, we think they will be extremely useful. First our study describes not only the current models but the method in general. The specific alterations will allow to understand how sarcoma with specific genetic alterations respond to particular drug treatments for example. Moreover, the same strategy described here can be used to model other alterations in the future and should be shared to the sarcoma research community.

I will restrict my comments to the models I took issue with in the original review

KRAS + p53. (now no longer described as ERMS). Despite renaming the tumor model, the

manuscript continues to imply that these are recapitulating anaplastic ERMS (as in the following sentence), “KRAS/sgTrp53 tumors, exhibited focal or diffuse anaplasia often observed in patient specimens of TP53-mutated RMS.” As noted in the revised manuscript and in the Reviewer’s Figure 2, these do not well recapitulate ERMS (regardless of anaplastic features) morphologically or any other sarcoma given negative desmin, myogenin, and nonspecific clustering with human vaguely myogenic tumors. Human sarcomas aside from ERMS rarely have KRAS mutations, although RAS pathway dysregulation may be seen by other mechanisms (for instance in MPNST, IFS, various kinase-altered sarcomas). While KRAS mutations are well known to cause sarcomas in mice, I have always regarded the relevance to human tumors to be dubious. This study does little to assuage this concern. Labeling the tumor only by its genotype does not make it more relevant to human disease, nor are KRAS mutant mouse models of sarcoma novel.

We acknowledge that KRAS-driven models of sarcoma in mice exhibit histological heterogeneity. However, some KRAS/sgTrp53 were positive for desmin and focal myogenin (Supplementary Fig. 4b and Fig3a).

We added the text below so that this is clear:

“KRAS/sgTrp53 tumors, exhibited focal or diffuse anaplasia often observed in patient specimens of TP53-mutated RMS³¹. Although some tumors were positive for desmin and myogenic markers, not all showed strong staining as seen in human RMS and the tumors did not fully recapitulate the histological features of eRMS.”

While KRAS mutations are not the predominant drivers in human rhabdomyosarcoma, RAS pathway activation is frequently observed, making these models relevant for studying disease mechanisms. Finally, while KRAS-driven sarcoma models are not perfect or novel per se, the main point is that the approach using electroporation provides an alternative and potentially more accessible method for generating these tumors compared to traditional engineered models.

My original critique of the PAX3::FOXO1 model was that it lacks strong/diffuse myogenin staining, a feature of human ARMS. The authors’ rebuttal notes that other published mouse models of PAX3::FOXO1 RMS also only show focal myogenin/myod1 and that their model clusters with ARMS in the transcriptomic analysis, and expressed high mRNA levels of myod1 and myog and exhibit high levels of FGFR4. In this instance, I will accept that this model may have value, but if other mouse models of ARMS exist, how does this one improve or add value to those?

We have been clear in the text that there are previous models for synovial sarcomas and alveolar rhabdomyosarcoma. Still, they much more complicated to the somatic approach described here, and efforts for create allograft models have been lacking. The fact that the electroporation is able to create tumors without germline alterations and conditional alleles offers the possibility.

KRAS – empty vector ERMS: The case depicted in 3b could be any undifferentiated spindle cell sarcoma. As mentioned above, I am unconvinced that KRAS mutant models of sarcoma

in mouse are a good model for human disease, and they are definitely not novel (albeit the existing ones are engineered through other means than electroporation).

Thank you for your feedback. We acknowledge that KRAS-driven models of sarcoma in mice can exhibit histological heterogeneity and the example in Figure 3b may resemble an undifferentiated spindle cell sarcoma. While KRAS mutations are not the predominant drivers in human rhabdomyosarcoma, RAS pathway activation is frequently observed, making these models relevant for studying disease mechanisms. Additionally, while KRAS-driven sarcoma models are not perfect or novel per se, the main point is that the approach using electroporation provides an alternative and potentially more accessible method for generating these tumors compared to traditional engineered models.

KRASG12V/Bcor. Again, the authors have dealt with nomenclature concerns by removing the descriptive label of ERMS. The question still remains, what model of human sarcoma is this tumor useful for? The original aim was to generate a model of the minor subset of ERMS with BCOR mutations, but this model is not that, and it does not seem to recapitulate any known human sarcoma type. This model may be novel, but I am not entirely convinced this will help to better understand human sarcomas, in its present form. The authors suggest that the value of the model is to show that bcor inactivation cooperates with KRAS in tumorigenesis, and that bcor inactivation allows oncogenesis in different cells, resulting in a different methylation pattern. I would think it just as likely that bcor inactivation results in epigenetic reprogramming, as the authors have not put forth clear evidence as to what specific electroporated cells are giving rise to any of these models.

Thank you for your detailed feedback. We acknowledge the challenges in directly aligning this model with a specific human sarcoma subtype. While our initial goal was to model the minor subset of ERMS with BCOR mutations, we recognize that the resulting tumors do not fully recapitulate this subset. However, we believe that the KRAS/Bcor model remains valuable in demonstrating how BCOR inactivation can cooperate with KRAS in tumorigenesis and potentially drive epigenetic reprogramming, as reflected in the distinct methylation patterns observed.

Regarding the origin of the tumor cells, we agree that a more precise characterization of the electroporated cell population would strengthen our conclusions. While our current data suggest that BCOR loss influences lineage commitment, further studies are needed to definitively determine the identity of the initiating cells. We appreciate this point and clarified these limitations in the discussion:

“Further studies are needed to determine which muscle-resident cells are targeted by electroporation and to mechanistically elucidate the role of BCOR loss of function in sarcoma and other pediatric tumors where it is frequently observed²⁸.”

The histology of NTRK-pleo tumors is nonspecific and could be any high grade undifferentiated sarcomas. By mRNA profiling, it clusters with other sgP53 tumors with a pleomorphic morphology (Reviewer’s Figure 2). However, none of these sgp53 tumor models clusters with human sarcomas most likely to show p53 mutations/loss (UPS, LMS,

MFS, MPNST). As a result I am not sure what to make of these tumor models. The revised text however, does acknowledge that the cell of origin may be deterministic in these mouse models, affecting how well they are able to recapitulate human tumors.

For the NTRK-pleomorphic tumors, we recognize in the text that their histology is nonspecific and that their clustering with other sgP53-driven tumors does not directly map onto human sarcomas with p53 mutations/loss, such as UPS, LMS, MFS, or MPNST. However, we believe this highlights an important consideration in sarcoma modeling—that the cell of origin plays a key role in shaping tumor phenotypes. As noted, we have revised the text to better acknowledge this point in the discussion:

“It is also important to note that the same genetic alteration could give rise to tumors with distinct histologic and molecular characteristics (e.g., *ETV6::NTRK3*-driven tumors). This heterogeneity suggests that, for certain oncogenic events, multiple cellular contexts may be permissive to transformation; however, not all resulting tumors will faithfully recapitulate the human disease. As such, histological and cross-species comparisons are critical to confirm that an EPO-GEMM—or any given model—accurately reflects its human counterpart.”

sgNF1/sgtrp53: The renamed model could potentially be a model for some types of human RMS, which do sometimes have NF1 loss, so I will accept this model could have value.

We agree. Thank you

Additional minor critiques

Line 267 “synchronous Smarcb1 inactivation, pathognomonic for Malignant Rhabdoid Tumors (MRT) and Epithelioid Sarcomas (EpS),” SMARCB1 activation, while present in MRT and epithelioid sarcomas, is widely seen in a variety of other malignancies, including carcinomas, a subset of myoepithelial tumors, epithelioid schwannomas and epithelioid MPNST, an unusual vulvar malignancy with yolk-sac-tumor-like differentiation, and many others. Smarcb1 deficiency may well accelerate tumor development in this model, but the phenotype of the tumor it is driving is unclear. I would probably tone down the text to say that the tumors with Smarcb1 inactivation share the epithelioid to rhabdoid histology typical of the various human tumors with this alteration, although the exact histotype seen here is unclear (as it honestly also is in many human SMARCB1 deficient tumors).

We agree we have edited the text accordingly:

I note that while the tumor models have been renamed as per their genetic features, the figures, (for example figure 2 b/d) very clearly attempt to draw parallels to human tumors with the same alterations (e.g. ERMS) which these lesions do not in fact recapitulate). I feel that this is somewhat misleading to the reader.

We added the figures for direct comparison as this would be the “expected outcome”. However, in the text we clearly say that the tumors:

“*KRAS/sgTrp53* tumors, exhibited focal or diffuse anaplasia often observed in patient specimens of *TP53*-mutated RMS³¹. Although some tumors were positive for desmin and myogenic markers, not all showed strong staining as seen in human RMS and the tumors did not fully recapitulate the histological features of eRMS.”

Lines 556-558 “Most importantly, the somatically induced sarcoma models were faithful representatives of previously established conventional GEMMs and the human disease. They reproduced entity-specific histologies with remarkable accuracy and activated expression programs that reflect the biology of the human disease” is only true for some of the models – the way this is phrased implies that all the models are accurate representations of human disease, which the authors have already acknowledged is not the case for the mutation-associated models described above. This language should be toned down or made more specific to the fusion models such as SS18::SSX, ASPCR1::TFE3.

We edited the text so that is clear this applies to fusion driven sarcomas:

“Most importantly, the somatically induced sarcoma models were faithful representatives of previously established conventional GEMMs and the human disease. Particularly for fusion-driven sarcomas, these models faithfully recapitulated entity-specific histologies and activated transcriptional programs that closely mirror the biology of the human sarcomas.

Overall, I do believe that this resource could be valuable. However, my suggestions would be to simplify the claims and descriptions of the various *kras/bcor/nf1/p53* / *ntrk-pleo* models to make the manuscript easier to follow. I think it would be reasonable to say that while the tumors showed imperfect correlation to specific human tumors, they may be used as a resource to better understand the role of these mutations in transforming precursor cells.

We hope that the edits will help clarify how some genotypes do not fully correlate with human tumors.

Thanks for your edits, we think that they helped clarify caveats in specific genotypes.